# Impact of economic globalisation on value-added agriculture, globally

**Nadeena Sansika** [1], **Raveesha Sandumini** [1], **Chamathka Kariyawasam** [1], **Tharushi Bandara** [1], **Krishantha Wisenthige** [1], **Ruwan Jayathilaka** [2] *

1 Sri Lanka Institute of Information Technology, SLIIT Business School, Malabe, Sri Lanka, 2 Department of Information Management, Sri Lanka Institute of Information Technology, SLIIT Business School, Malabe, Sri Lanka

* ruwan.j@sliit.lk

**Data Availability Statement:** All relevant data are within the paper and its Supporting Information files.

**Funding:** The authors received no specific funding for this work.

## Abstract

Economic globalisation is the integration of national economies into the global economy through the increasing flow of goods, services, capital, and technology across borders and it has contributed to garnering a significant portion of most nations' national income, although its agricultural value-added aspect has yet to be maximised. This pioneering study explores the impact of economic globalisation on value-added agriculture in a global context based on countries' income levels. Panel data regression with the stepwise method was employed to quantify the impact of economic globalization on agriculture value added in 101 countries between 2000 and 2021. The findings of our study reveal that economic globalisation, through various channels such as fertilizer consumption, employment in agriculture, agriculture raw materials export and import, exchange rate, and foreign direct investment, significantly influences the agricultural value-added factor globally and across different income levels. Furthermore, the results show that agricultural employment significantly impacts the agricultural value-added factor globally and across all income levels. Also, countries with low and lower-middle-income levels significantly affect agricultural value-added due to exchange rates. In comparison, high-income and lower-middle-income levels have an impact due to foreign direct investment. Finally, the upper-middle-income countries have significantly affected agricultural value-added due to agricultural raw materials imports. This study confirms that employment in agriculture, exchange rate and foreign direct investments positively impact agriculture value-added on the global level and based on the income level of countries.

## Introduction

Economic Globalisation (EG) has significantly impacted Agriculture Value Addition (AVA), with studies showing positive and negative effects [1] On the positive side, globalisation has increased access to markets for agricultural products, leading to more significant trade and economic growth. Furthermore, this situation has allowed farmers to benefit from higher crop prices and access to modern technologies and innovations that can improve productivity [2].

**Competing interests:** The authors have declared that no competing interests exist.

The concept of EG is a historical process that resulted from human ingenuity and technological advancements, has simplified this process and AVA describes how a sector's net output is calculated by summing up all the results, deducting intermediate inputs, and adjusting for agronomy, forestry, hunting, fishing, livestock production [3,4]. In addition, globalisation can transform rural agriculture into more commercialised and value-based agriculture and improve the rural community's living conditions [5]. The contemporary era of globalization recognizes the significance of financial and natural resources as crucial factors that play a vital role in reducing environmental degradation, while simultaneously promoting economic growth [6].

Extant empirical literature investigates how AVA in BRICS-T (Brazil, Russia, India, China, South Africa, and Turkey strengthens the potential for the region's ecological footprint to increase and how a one Per cent influence on agriculture raises it by 0.2201 Per cent [7]. Furthermore, Past studies indicate that factors such as AVA, economic growth, non-renewable energy use, and tourism sector expansion have a significant impact on environmental degradation, highlighting their adverse effects on the quality of the environment [8]. Trade is crucial in the global agricultural sector. Several interrelated trade theories impact AVA. Two of the most influential theories are the Heckscher-Ohlin and New Trade theories. The Heckscher-Ohlin theory suggests that countries will specialise in producing goods that use their abundant factors of production. In contrast, the New Trade theory suggests that economies of scale and increasing returns to specialisation can create a comparative advantage which leads to trade. Both theories can lead to increased value addition in agriculture as countries and firms specialise in producing goods where they have a comparative advantage and invest in research and development to improve the quality of their products. In summary, international trade can increase value addition in agriculture [9]. Furthermore, Trade has received exceptional attention since agricultural production is globalised through agriculture and is the source of globalisation's advantages. It proposes that low-income countries should have access to high-income countries' markets, especially for high-value crops [10]

Foreign direct investment (FDI) is a significant factor in determining the level of agricultural production in a country, as it brings in new technologies and skills that can benefit farmers and improve the overall productivity of the agricultural sector [11]. FDI has demonstrated to be a significant factor in fostering agricultural growth by providing a source of outside capital in developing nations. According to a performance index in China, agriculture is receiving more foreign FDI, but not at a satisfactory rate given the industry field size [12,13]. Moreover, a positive correlation between the real GDP growth rate and the ratio of FDI inflows to value added to GDP was demonstrated [13].

The Exchange rate (ER) can also impact AVA, specifically through ER and terms of trade. Two exchange theories that are particularly relevant in this context are the Marshall-Lerner condition and the Prebisch-Singer hypothesis. The Marshall-Lerner condition theory states that a depreciation of a country's currency will improve its trade balance if the sum of price elasticities of exports and imports, is greater than one. In agriculture, a country's currency depreciation can make its exports cheaper and more competitive on the international market, leading to an increase in demand and value -addition. Furthermore, the Prebisch-Singer hypothesis theory suggests that the terms of trade of primary commodity exporters (including many agricultural commodities) tend to deteriorate over time, as the prices of manufactured goods that they import tend to rise faster than the prices of the commodities they export. This can lead to a decline in the value of agricultural exports and may reduce incentives for value addition [14,15].

Changes in ER and macroeconomic policies may be one of the factors influencing agricultural prices, and these changes significantly impact Canadian AVA [16]. Further findings

demonstrate that Employment in Agriculture (EA) and Fertilizer Consumption (FC) affect AVA. EA mean working-age persons who engaged in any activity to produce agricultural goods or provide services for money or profit [4]. Agriculture is vital for a country's economy, and enhancing the embedded location of agricultural value chains is crucial for its modernisation. The global division of labour has led to an increase in the role of the agricultural global value chain, shifting agricultural trade from single-country production to multi-country production. This has refined the international division of labour in agriculture and extended production chains, leading to national agriculture being included in the global division of labour system dominated by multinational corporations [17]. Prior investigations found a positive association between FC and GHG emissions, indicating a need to avoid excessive use of fertilizers and pesticides in sustainable agriculture. The government should impose restrictions on the use of chemical fertilizers and engage in research and development to develop environmentally sustainable fertilizers and new crops that do not rely on hazardous fertilisers. Organic and low-carbon agriculture systems should be encouraged to reduce emissions and improve carbon sequestration. The study suggests policymakers promote organic farming, tunnel farming, no-till farming, and limit fertilizer use to reduce environmental impact [18]. In addition, In Ethiopia, the use of fertilisers positively impacts yield value and agricultural production [19].

Agriculture Raw Material Exports (ARME) and Agriculture Raw Material Imports (ARMI) significantly affect the foreign currency inflows and outflows of nations. As these products often constitute a significant share of many countries' exports and imports, understanding their effects on foreign currency is important for policymakers and stakeholders in the agriculture sector. Exports can drive economic growth by enabling a country to purchase essential capital goods, technology, and manufactured goods. A country with abundant natural resources that can be used as raw materials can benefit from increased exports and higher AVA per GDP. Meanwhile, ARMI can enhance agricultural productivity, reduce production costs, and guarantee food security, indirectly increasing ARMI [4,20].

This study allows for a more nuanced understanding of how globalization affects different actors in the agricultural supply chain and this study important in three ways. Firstly, Value-added agriculture can have a significant impact on income levels, where agriculture is often a major source of employment and income. By adding value to agricultural products, farmers and other actors in the supply chain can increase their profits and improve their standard of living.

Secondly, at the same time, EG can have both positive and negative effects on value-added agriculture and income levels. On the positive side, globalisation can create new opportunities for agricultural producers to access global markets and increase their profits. This can lead to increased incomes and improved living standards for those involved in value-added agriculture. On the negative side, globalisation can expose smaller producers to increased competition from larger, more efficient producers in other countries. This can lead to lower prices and reduced profits, which can hurt incomes.

Thirdly, research examining EG's impact on value-added agriculture and income levels separately can help policymakers and other stakeholders better understand the trade-offs associated with global integration. It can also inform the development of strategies to maximise the benefits of globalisation, while minimising its negative impacts.

The main objective of this study is to investigate whether EG has impacted AVA at the global level and income levels between 2000–2021. To achieve this objective, the study aims to answer the following research questions:

1. Does EG impact AVA at the global level between 2000–2021?

2. How does the impact of EG on AVA vary by income levels of countries between 2000–2021?

3. What are the comparative results of the impact of EG on AVA at both global level and different income levels?

By addressing these research questions, the impact of EG on AVA is a critical and underexplored area of research. This study fills this gap by conducting a comprehensive analysis for both globally, and income levels: high-income, lower-middle-income, upper-middle-income, and low-income countries. By examining the impact between EG and AVA across different income levels, the study provides valuable insights for policymakers, researchers, and stakeholders seeking to enhance agricultural productivity and income generation. This research significantly contributes to the existing literature and offers a solid foundation for evidence-based decision-making.

This research was conducted globally with four different income levels: high-income, lower- and upper-middle-income countries based on World Bank categorization. Authors have considered the period from 2000 to 2021 based on data availability for all variables selected. The study highlights crucial factors addressing the research gaps, thus rendering it unique. First, the study was conducted separately to analyse the four global income levels, using data from 101 nations. This exercise allows a more comprehensive understanding of the impact of EG on AVA at different income levels. Second, the study will apply a novel evaluation of the stepwise approach using panel data regression. This approach provides a better understanding of the impact of EG on AVA over time and how this impact changes based on the income level of a country. Last, the variables and time frames selected for the analysis differ from those used in previous studies. Therefore, it will provide a fresh perspective on the topic and help fill gaps in the existing research.

The paper is structured into five sections. The first section is the introduction, followed by a summary of related literature in the second section. The third section outlines the data and methods, including an overview of the dependent and independent variables globally and by income level. The fourth section interprets the results and discussion. Finally, the fifth section outlines policy implication, future research, limitation and conclusion.

## Literature review

The impact of Economic Globalization (EG) on Agricultural Value Addition (AVA) has been the topic of substantial research, which has shown that EG has both good and negative effects on AVA. As a result, many nations are presently striving to improve their economic competitiveness in the age of globalization by utilizing a variety of industries, including agriculture, which has numerous benefits [21]. Many past researchers have demonstrated that EG negatively and positively influences AVA. EG is one factor that primarily determines the type of AVA output produced in a country.

It alludes to the growing economic integration of nations, primarily due to cross-border trade [22]. Adding value is changing or transforming a product from its original state into a more valuable form preferred in the marketplace. The expansion of international trade is a critical component of globalisation [23]. Today, EG is used in various ways, including export and import, the ER, and FDI [24]. Another study explores the relationship between EG, AVA, and the ecological footprint in the E7 nations: Brazil, China, India, Indonesia, Mexico, Russia and Turkey. The results indicate that these factors have contributed to environmental deterioration, and policymakers should implement environmental damage costs and maintain strategic resource control measures for a sustainable environment [25].

While studies examining the impact of EG on AVA exist, there is limited research on how this impact differs across income levels. Understanding the impact of EG on AVA across income levels is essential because the impact of EG on AVA may be different for all income levels. This literature review aims to fill this gap by examining the impacts of various factors contributing to EG on AVA across income levels and provide a comprehensive understanding of the relationship between EG and AVA for different income categories. This review will contribute to the current body of literature by providing insights into the impact of EG on AVA for different income levels, which can inform policies promoting economic growth.

This section thoroughly examines the literature from previous studies on the impact of EG on AVA at the global level.

## Global level

In any income level, the EG factor supports the integration of AVA globally. According to World Bank international trade expansion is a vital aspect of globalization, and it has been observed that both exchange rates (ER) and foreign direct investment (FDI) exert a similar influence [23]. AVA fluctuations have multiple significant factors contributing to them, with one of the key factors being changes in exchange rates (ER) [16,26–28]. Various studies have indicated that regardless of a country's income level, exchange rates have both positive and negative impacts on AVA [29].

The potential benefits of Foreign Direct Investment (FDI) on Agricultural Value Added (AVA) have been recognised, offering opportunities to enhance the rural economy and provide economic advantages to farmers [13,22]. Research conducted by Coltrain, Barton indicates a positive correlation between the rate of real GDP growth and the ratio of FDI inflows to value added to GDP [22]. However, the OECD suggests that FDI performance in China's AVA is still unsatisfactory given the market scale, presenting an alternative viewpoint in the literature [30].

Moreover, numerous researchers contribute to the field of study through a variety of findings. For example, cooperatives with added value can form new alliances by expanding employment opportunities. Most of the time, the connection flows from AVA per worker to GDP per capita [22]. The reduction of Agricultural Value Added (AVA) is observed as a result of the labour force and trade openness [31]. However, in some cases, this reduction in labour in developed countries is primarily made feasible by the enormous productivity improvement [32].

According to past studies, FC plays a significant role in increasing AVA [33,34]. These studies suggest that boosting used fertiliser is linked to an increase in AVA. One study find that the use of fertilizers has a significant negative impact on AVA growth [35]. They suggest that G8 countries could reduce their FC. One study indicate that the agricultural performance of identified secondary raw material-derived fertilizers supports their use in both organic and conventional European agricultural sectors [36]. Previous investigations revealed that labour force participation harms economic growth in Southern Asia, but it has a positive effect in Western Asia. Moreover, the study found a robust and positive relationship between trade transparency, human capital, and economic growth [37]. ARME significantly impacts AVA, as increased exports lead to increased revenue and higher value-added. The World Bank found that increased ARME is positively associated with economic growth and higher AVA [38]. On the other hand, increased ARMI can also positively impact AVA by reducing production costs and increasing competitiveness, leading to higher value-added. ARMI positively affects AVA in the agriculture sector [39].

Economic Globalization factors such as international trade, exchange rate, foreign direct investment, fertiliser consumption, and agricultural raw material exports have significant impacts on agriculture value addition (AVA) at the global level. Despite some mixed findings, the literature suggests that these factors can positively influence AVA, leading to increased revenue and economic growth. However, it is also essential to consider the potential negative impacts, such as the negative effects of fertiliser consumption on AVA. Therefore, policymakers need to adopt a holistic approach to economic globalization to promote sustainable agricultural development and enhance AVA.

## High-income

This section will elucidate how researchers have examined the issue from past literature to determine how EG affects AVA at the high-income level. Due to the slow growth or actual reduction in spending on agricultural productivity-enhancing studies in high-income countries, countries having similar studies have seen a break in their AVA growth. However, the pattern of AVA worldwide has changed significantly in recent decades due to uneven AVA growth over time and between different countries, raising the prospect of more changes in the years to come [28]. An essential link between producing agriculture and the AVA in the United States economy and AVA goods have over $700 billion in annual retail sales, employing over 20% of the labour force in the United States [40].

Another element of EG is ER. The long-run effect shows that the conditional effect of the ER on the relationship between oil income and AVA is negative and statistically significant at the conventional level for the entire sample [28]. According to the study's findings, the ER significantly impacts Canadian agriculture. The relative prices between the agricultural and non-agricultural sectors of the economy would shift due to the dollar's depreciation [26].

EG factors such as international trade, FDI, ARME, FC, and ER have significant impacts on AVA in high-income countries. Despite the slow growth or actual reduction in spending on agricultural productivity-enhancing studies in high-income countries, AVA remains a crucial component of their economies, providing significant retail sales and employment opportunities. However, the pattern of AVA growth has been uneven over time and between different countries, highlighting the need for further research and development in this area. Therefore, understanding how EG factors affect AVA is of utmost importance for policymakers and stakeholders to promote sustainable economic growth and development in high-income countries.

## Low-income level

According to previous studies, EG can influence AVA in low-income countries to help those nations rise above their economic situation. In developing countries, FDI inflows and ARME raise AVA [21]. However, it has been demonstrated by Persson that Ethiopia's investment policy for attracting foreign direct investment (FDI) in large-scale agriculture is inadequate. The policy primarily emphasizes providing incentives to attract FDI rather than ensuring availability and support [41].

Numerous prior studies have provided conflicting evidence to support the literature on how EG effect AVA. Another crucial group of EG variables that influence AVA are ER and EA. Some studies have demonstrated that rising ER leads to depreciation in Africa's value-added agriculture [34,42]. On the other hand other studies have found that various EG factors, including ER and EA, have an influence on AVA [43]. Using a unidirectional causation flow from AVA to market capitalisation and stock value exchanged, it has been demonstrated that bidirectional causality can be established between labour and AVA in Africa [44]. For many

African and West Asian countries, it is crucial to increase the use of chemical pesticides and fertilisers through more value-based manufacturing [45]. Similarly, it has been claimed that ARMI in Tunisia and Egypt have a long-term positive effect on value-based agricultural growth [46].

EG factors such as FC, EA, ARME, ARMI, Trade, ER, and FDI on AVA in low-income countries. However, conflicting evidence also exists on the impact of EG on AVA. Understanding the impact of EG on AVA in low-income countries is essential as it can help these nations rise above their economic situation and improve their agricultural productivity.

## Lower middle-income level

This section will examine the findings in the literature regarding how EG impacts AVA in Lower-middle-income countries. According to past researchers, FDI has medium and long-term positive effects on AVA in lower-middle-income countries. In many developing countries, FDI has a positive impacts value-added in agriculture by facilitating technological advancements and generating new employment opportunities [11,47].

One study suggests that governments should take high level initiatives to promote Foreign Direct Investments (FDI) and international trade in Asian countries. It recommends engaging more in economic organisations, enacting progressive legislation, and encouraging interaction with world economies [48].

Many past findings claim that there is a positive impact of EG factors like ER, ARMI, ARME, and EA on AVA in lower-middle-income countries. For example, it has been demonstrated that the labour force and Agricultural Resource Management Efficiency (ARME) have a positive and significant impact on Malaysia's high rice output [49]. The literature is supported by researches who demonstrate the favourable and considerable impact of FDI, the real effective ER, ARMI, and ARME on AVA [50,51]. Moreover, they contend that a rise in investment in the industry benefits agricultural output.

A further study reveals some divergent viewpoints from earlier researchers. AVA is being reduced as a result of the labour force and trade openness in developing countries [31], while AVA growth is linearly related to the proportion of women in the agricultural labour force, whether the developing country is agriculturally based, and whether it is in Europe or Central Asia [52]. A previous researchers examined the role of human capital in promoting sustainable development and reducing carbon emissions [53]. The findings reveal that investing in environmentally friendly research and development can help lower carbon emissions and increase human capital. However, the study suggests that a green development strategy can only materialise if the government invests more in education, healthcare, and improvements to the employment market.

In a nation like India, value-based agricultural growth is necessary due to their agricultural economy, so the agricultural worker productivity needs to be adequately addressed on a time series basis to determine the worker's marginal productivity [54]. Also, the fundamental change in the Indian value-based agricultural sector, represented by the increased emphasis on exports, has been highlighted [55]. It is transitioning from labour-based agriculture to market-based agriculture. Another study suggested that Vietnam should enhance AVA by introducing modern agro-based technology and promoting sustainable agriculture, such as low-carbon agriculture systems and the use of renewable energy, and by avoiding excessive use of fertilizers and pesticides [56].

In lower-middle-income countries, previous research indicates that FDI has a positive impact on AVA by creating technological advancements and new job opportunities. Additionally, factors such as ER, ARMI, ARME, and EA have been shown to have a favourable and

considerable impact on AVA. However, there are some divergent viewpoints regarding the impact of trade openness and the labour force on AVA. It is important for these countries to focus on value-based agricultural growth and invest in the industry to benefit agricultural output. The findings suggest that EG can play a significant role in promoting AVA in lower-middle-income countries, leading to economic growth and job creation.

## Upper middle-income level

A few earlier studies examined how EG impacted AVA at the upper-middle-income level. By adopting panel data regression, the impact of EG on AVA in 17 developing nations from 2006–2018 was analysed [21]. It was observed that the value of agricultural exports and FDI inflows considerably impact AVA. In this instance, EG has been demonstrated to activate AVA. This outcome is because AVA is globalised through foreign agricultural trade and foreign investment in the agricultural sector [57].

A connection between financial performance and agriculture was found in a Malaysian study [58]. The study's findings indicate that exported processed agro-products, or AVA, significantly impact a company's financial performance, while ARME has little to no effect. In another study from Ecuador, it was argued that the improvement in relative AVA was achieved through an appreciation of the ER [59]. The response of agriculture to the improvement in the ER regime is the only bright spot in the economy of the 1980s.

EG factors like FDI inflows and agricultural exports significantly impact AVA in upper-middle-income countries, as shown in previous studies. However, the impact of other EG factors like ARME on AVA is less significant. Appreciation of the ER has also been shown to improve relative AVA in some countries. Understanding how EG factors impact AVA is crucial for policymakers to formulate effective strategies to promote agricultural growth and improve the overall economic situation in these countries.

Panel data regression combines cross-sectional and time-series data to analyse changes in variables within and between entities over time. Cross-panel analysis is a type of panel data analysis that compares variable interdependence across different nations, allowing researchers to evaluate policy effects and identify relationships between variables across various periods and cross-sections. Cross-panel regression provides valuable insights into the global impact of policies by exploring the simultaneous movement of both dependent and independent variables. The utilization of the cross-panel technique is also viable for conducting research on this area [60].

This study aims to investigate the impact of various economic factors on AVA within specific income levels and globally. Based on a comprehensive literature review, the study examines hypotheses relating to the effects of FC, EA, ARME, ARMI, Trade, ER, and FDI on AVA, where there is a significant impact of EG having an impact on AVA globally and high- income, low- income, lower -middle income and upper -middle income separately between 2000–2021.

Almost all the literature review above has focused on separate aspect AVA. Therefore, there arises a need to fill this literature gap.

## Data and methodology

This study was reviewed and approved by Sri Lanka Institute of Information Technology (SLIIT) Business School and the SLIIT ethical review board. Study used the secondary data sources and the data file used for the study is presented in S1 Appendix. The data analysis was done using panel data regression, which included observations regarding various cross-sections. The stepwise method was used to establish the final model identification.

**Table 1. Data sources and variables.**

| Variable | Definition | Measure | Source |
|---|---|---|---|
| **Dependent Variable** | | | |
| **AVA** | Agriculture, forestry, and fishing, value-added | (% of GDP) | The World Bank https://data.worldbank.org/indicator/NV.AGR.TOTL.ZS |
| **Independent Variables (Economic Globalisation)** | | | |
| **FC** | Fertiliser Consumption | (Kilograms per hectare of arable land) | The World Bank https://data.worldbank.org/indicator/AG.CON.FERT.ZS |
| **EA** | Employment in Agriculture | (% of total employment) (modelled estimate) | The World Bank https://data.worldbank.org/indicator/SL.AGR.EMPL.ZS |
| **ARME** | Agricultural Raw Materials Exports | (% of merchandise exports) | The World Bank https://data.worldbank.org/indicator/TX.VAL.AGRI.ZS.UN |
| **ARMI** | Agricultural Raw Materials Imports | (% of Merchandise imports) | The World Bank https://data.worldbank.org/indicator/TM.VAL.AGRI.ZS.UN |
| **Trade** | Trade | (% of GDP) | The World Bank https://data.worldbank.org/indicator/NE.TRD.GNFS.ZS |
| **ER** | Exchange Rate | (LCU per US$, period average) | The World Bank https://data.worldbank.org/indicator/PA.NUS.FCRF |
| **FDI** | Foreign Direct Investment | (Net inflows % of GDP) | The World Bank https://data.worldbank.org/indicator/BX.KLT.DINV.WD.GD.ZS |

Source: Compiled by authors.

The authors have used four income level categorisations of countries by the World Bank based on their Gross National Income (GNI) per capita. Low-income countries: with a GNI per capita of $1,045 or less. Lower-middle-income countries: with a GNI per capita between $1,046 and $4,125. Upper-middle-income countries have a GNI per capita of between $4,126 and $12,735. High-income countries: with a GNI per capita of $12,736 or more [61]. Data from 101 countries, comprising 32 high-income countries, 11 low-income countries, 28 lower-middle-income countries, and 30 upper-middle-income countries, were collected to examine the study's aims. The data period covered from 2000 to 2021 based on the availability of data for all variables selected for the study. Secondary data gathering from reliable sources is anticipated to be used to investigate the impact of EG on AVA at the global level and income levels., Stata statistical software was used to analyse the data. Data were collected under AVA, FC, EA, ARME, ARMI, Trade, ER, and FDI. Table 1 represents data sources and variables. The data file used for the analysis can be found in Appendix.

The stepwise method adds or removes predictor variables in a regression model based on statistical criteria such as p-values, t-statistics, or F-statistics. The stepwise approach aims to find the best subset of predictor variables that provides the most accurate and parsimonious explanation of the relationship between the dependent and predictor variables [62].

The mathematical model presented in this study comprises a comprehensive set of variables as outlined in Eq 1. This equation serves as the foundation for the development of further Eqs 2–6, which have been derived using a stepwise method.

$$AVA_{it} = \beta_0 + \beta_1(FC_{it}) + \beta_2(EA_{it}) + \beta_3(ARME_{it}) + \beta_4(ARMI_{it}) + \beta_5(Trade_{it}) + \beta_6(ER_{it}) + \beta_7(FDI_{it}) + \varepsilon_{it} \qquad (1)$$

In this Eq 1, $AVA_{it}$ represents the value of the dependent variable at time t and i counties, and $\varepsilon_{it}$ represents the residual error term for time t. The coefficients $\beta_0$, $\beta_1$, $\beta_2$, $\beta_3$, $\beta_4$, $\beta_5$, $\beta_6$, and $\beta_7$ represent the intercept and slopes of the regression line, which describe the impact of

the independent variables on the dependent variable $\text{AVA}_{it}$.

$$\text{AVA}_{it} = \alpha_0 + \alpha_1(\text{EA}_{it}) + \alpha_2(\text{ARMI}_{it}) + \alpha_3(\text{FC}_{it}) + \alpha_4(\text{FDI}_{it}) + \alpha_5(\text{ER}_{it}) + \varepsilon_{it} \tag{2}$$

At a global level, Eq 2 has been established. The equation models the impact of five independent variables on the dependent variable $\text{AVA}_{it}$ at time t and i counties. $\text{AVA}_{it}$ represents the value of the dependent variable for each period. At the same time, $\varepsilon_{it}$ is the residual error term that captures the difference between the observed value of the dependent variable and the predicted value-based on the independent variables. The coefficients $\alpha_0$, $\alpha_1$, $\alpha_2$, $\alpha_3$, $\alpha_4$ and $\alpha_5$, represent the intercept and slopes of the regression line.

$$\text{AVA}_{it} = \phi_0 + \phi_1(\text{EA}_{it}) + \phi_2(\text{FDI}_{it}) + \phi_3(\text{ER}_{it}) + \phi_4(\text{ARMI}_{it}) + \phi_5(\text{FC}_{it}) + \varepsilon_{it} \tag{3}$$

Eq 3 has been established based on high-income-level analysis. $\text{AVA}_{it}$ represents the value of the dependent variable for each period, while $\varepsilon_{it}$ is the residual error term. The intercept $\phi_0$ represents the expected value of the dependent variable when all independent variables are equal to zero. The slopes $\phi_1$, $\phi_2$, $\phi_3$, $\phi_4$, and $\phi_5$ represent the change in the dependent variable associated with a unit change in each independent variable while holding all other independent variables constant.

$$\text{AVA}_{it} = \delta_0 + \delta_1(\text{FC}_{it}) + \delta_2(\text{EA}_{it}) + \delta_3(\text{ER}_{it}) + \delta_4(\text{ARMI}_{it}) + \delta_5(\text{FDI}_{it}) + \varepsilon_{it} \tag{4}$$

In essence, Eq 4 provides a mathematical representation of the impact of independent variables on the dependent variable at the low-income level. $\varepsilon_{it}$ is the residual error term. The coefficients $\delta_0$, $\delta_1$, $\delta_2$, $\delta_3$, $\delta_4$, and $\delta_5$ represent the intercept and slopes of the regression line.

$$\text{AVA}_{it} = \gamma_0 + \gamma_1(\text{EA}_{it}) + \gamma_2(\text{FDI}_{it}) + \gamma_3(\text{FC}_{it}) + \gamma_4(\text{ARME}_{it}) + \gamma_5(\text{ER}_{it}) + \varepsilon_{it} \tag{5}$$

Eq 5 models the impact of independent variables on the dependent variable at the lower middle-income level. The residual error term, $\varepsilon_{it}$, captures any discrepancy between the observed value of the dependent variable and the value predicted by the independent variables. The coefficients $\gamma_0$, $\gamma_1$, $\gamma_2$, $\gamma_3$, $\gamma_4$, $\gamma_5$ represent the intercept and slopes of the regression line in the equation.

$$\text{AVA}_{it} = \theta_0 + \theta_1(\text{EA}_{it}) + \theta_2(\text{ARME}_{it}) + \theta_3(\text{ARMI}_{it}) + \theta_4(\text{ER}_{it}) + \theta_5(\text{FDI}_{it}) + \theta_6(\text{FC}_{it}) + \varepsilon_{it} \tag{6}$$

In simpler terms, Eq 6 provides a mathematical representation of the impact of independent variables on the dependent variable $\text{AVA}_{it}$ for each period at the upper middle-income level. It captures the expected value of $\text{AVA}_{it}$ and the effect of each independent variable on it. $\varepsilon_{it}$ is the residual error term, and $\theta_0$, $\theta_1$, $\theta_2$, $\theta_3$, $\theta_4$, $\theta_5$ represent the intercept and slopes of the regression line in the equation.

Multicollinearity is a statistical issue when the independent variables in a regression model are highly correlated. This can lead to consistent and reliable results and difficulties in interpreting the individual effects of each independent variable [63].

The Correlation metric was utilised to assess the presence of multicollinearity among the variables in all countries. In Eqs 2–4, the Trade variable was omitted due to its high correlation with other variables. The study employed the stepwise method to perform an analysis aimed at identifying the most appropriate variables for the final models, both globally and for each income level individually. The procedure involved executing panel data regression models using both fixed effects model (FEM) and random effects model (REM), which were then used to compute the t and z values for each variable. These values were sorted separately in descending order based on their magnitudes. Subsequently, the panel data regression models were re-

run for both FEM and REM using the sorted variables in descending order, one by one. If a variable's coefficient sign differed from previous literature findings, it was removed from the model, and the process was repeated until the best model was obtained for each income level and globally [64].

In the panel data regression, the ARME variable was removed for all countries, the high-income and low-income levels, the ARMI and Trade variables were removed for the lower middle-income level, and the trade variable was released for the upper middle-income level due to changes in the sign of the coefficient values. Table 4 includes the final models selected stepwise for all countries and income levels. Certain countries were excluded from the data file due to the absence of data for several variables. The missing values for the following variables were filled in using the "ipolate" and." "epolate" functions in the Stata software: 2020 and 2021 in the EA variable for all countries, 2021 in the FC variable for all nations, and 2020 and 2021 in the Trade variable for Burkina Faso and the AVA, ARME, and ARMI variables for some countries. No missing values were present in the FDI and ER variables.

A series of tests were used to determine the most suitable model for the analysis. These tests include the F test, the Breusch-Pagan Lagrange Multiplier (LM) test, and the Hausman test. The F test is a statistical test used to determine the overall significance of a model. It is used to evaluate the null hypothesis that all regression coefficients are equal to zero. In this case, the Pooled Ordinary Least Squares (POLS) and FEM were used to determine the best method through the F test. The Breusch-Pagan LM test is used to detect heteroscedasticity, which occurs when the variance of the errors is not constant. This test determines the best method between the POLS and the REM. Finally, the Hausman test determines whether a FEM or REM is more appropriate for the data. The Hausman test was performed to choose the most appropriate method between the FEM and REM [62,65]. The results of the specification tests are shown in S2 Appendix. The panel regression models employed to analyse the different income group effects of FC, EA, ARME, ARMI, FDI, ER on AVA are portrayed in S3 Appendix. The fixed and random effect estimates for the final stepwise model are presented in S4 Appendix.

In conclusion, these tests will provide crucial information on the suitability of the various models and help to determine the best approach for the analysis.

## Results

Descriptive statistics of variables for different income groups of countries are provided in Table 2, including all nations' lower-middle-income, and upper-middle-income. The variables are AVA FC, EA ARME, ARMI Trade, FDI and ER, for each variable and income group, the tab gives the number of observations, mean, standard deviation, and maximum value. These statistics provide a summary of the distribution of the variable and give an idea of the range and central tendency of the data. The information can be used to understand the distribution of the variables across countries with different income levels and identify patterns and trends in the data.

There are 2222 observations included here of which 704, 242, 616, and 660 comments correspond to high, low, middle, and middle income level countries, respectively, from year 2000 to 2021. In this case, the highest mean value (771.8038) comes from the ER variable for all countries. Additionally, the category with the highest mean value, 15.78708, belongs to the income level. Compared to the other income levels, the income level has the lowest ARME, while the low-income level has the greatest ARME.

In conclusion, Table 2 provides a comprehensive overview of the descriptive statistics of the variables for different income groups of countries, providing a clear picture of the distribution

**Table 2. Summary descriptive statistics for the key variables.**

| Countries | | Variables | | | | | | | |
|---|---|---|---|---|---|---|---|---|---|
| | | AVA | FC | EA | ARME | ARMI | Trade | FDI | ER |
| **All Countries** | **Obs** | 2222 | 2222 | 2222 | 2222 | 2222 | 2222 | 2222 | 2222 |
| | **Mean** | 10.3593 | 236.5969 | 24.9387 | 3.7590 | 1.4482 | 83.7457 | 4.9790 | 771.8038 |
| | **SD** | 9.3747 | 950.1158 | 21.9032 | 7.6511 | 1.0101 | 55.3287 | 16.3300 | 3222.375 |
| | **Min** | 0.0301 | -3147.054 | -0.03 | -0.4269 | -0.7374 | 16.3521 | - 40.0866 | -0.0973 |
| | **Max** | 44.1070 | 19171.85 | 91.76 | 75.4446 | 18.4825 | 442.62 | 449.0809 | 42000 |
| **High-Income** | **Obs** | 704 | 704 | 704 | 704 | 704 | 704 | 704 | 704 |
| | **Mean** | 2.4302 | 465.1768 | 5.6809 | 2.6101 | 1.1438 | 108.6091 | 7.7095 | 36.8647 |
| | **SD** | 2.0866 | 1624.154 | 6.2693 | 3.9529 | 0.7212 | 80.7301 | 28.1096 | 113.7587 |
| | **Min** | 0.0301 | -3147.054 | -0.03 | 5.33e-06 | 0.0077 | 19.5596 | - 40.0866 | -0.0973 |
| | **Max** | 13.1496 | 19171.85 | 45.21 | 30.0417 | 4.7909 | 442.62 | 449.0809 | 792.7272 |
| Low-Income | **Obs** | 242 | 242 | 242 | 242 | 242 | 242 | 242 | 242 |
| | **Mean** | 27.1758 | 10.7511 | 64.2367 | 8.3126 | 1.4122 | 54.5178 | 4.3038 | 833.1056 |
| | **SD** | 8.2229 | 14.2553 | 18.2090 | 12.6347 | 1.4259 | 19.8256 | 5.7595 | 912.6552 |
| | **Min** | 2.8607 | -2.6593 | 24.47 | -0.4269 | 0.1706 | 20.9640 | -3.7163 | 3.1108 |
| | **Max** | 44.1070 | 91.9524 | 91.76 | 75.4446 | 18.4825 | 127.2042 | 39.4562 | 3829.978 |
| **Lower Middle Income** | **Obs** | 616 | 616 | 616 | 616 | 616 | 616 | 616 | 616 |
| | **Mean** | 15.7870 | 99.1990 | 36.2600 | 5.0351 | 1.8796 | 69.8563 | 2.9401 | 2097.291 |
| | **SD** | 7.2675 | 126.4026 | 14.8031 | 10.6165 | 1.1655 | 30.1311 | 2.9786 | 5775.558 |
| | **Min** | 1.4001 | -1.3846 | 9.0400 | 0.0043 | -0.3667 | 16.3521 | -5.1603 | 0.5449 |
| | **Max** | 37.9524 | 600.078 | 82.99 | 74.8808 | 8.4946 | 186.4682 | 17.1312 | 42000 |
| **Upper-Middle-Income** | **Obs** | 660 | 660 | 660 | 660 | 660 | 660 | 660 | 660 |
| | **Mean** | 7.5852 | 203.8266 | 20.5046 | 2.1237 | 1.3835 | 80.9051 | 4.2171 | 296.1405 |
| | **SD** | 3.3586 | 346.6173 | 12.0997 | 1.9586 | 0.7714 | 34.8028 | 4.7784 | 1048.89 |
| | **Min** | 1.9268 | -18.0528 | -0.02 | 0.0427 | -0.7374 | 21.8522 | -5.0882 | 0.0876 |
| | **Max** | 25.4093 | 2299.422 | 55.3 | 11.5683 | 4.8079 | 220.4068 | 55.0703 | 6774.163 |

Note: Obs., Mean, SD, Min. and Max. Represent Observations, Standard Deviation, Minimum, and Maximum, respectively. Source: Authors' calculation based on data from the world bank.

and central tendency of the data. The results give us valuable insights into the patterns and trends of the variables across different income levels, enabling us to make logical comparisons and understand the relationship between income and the dependent and independent variables. The information presented in this table serves as a helpful reference point for further analysis and decision-making in economics.

Below, these figures present the data based on the average variations of the dependent and independent variables about income levels from 2000 to 2021. Fig 1 shows that high-income countries exhibit a lower average percentage of AVA than low-income countries. The lower-middle-income level has a higher, moderate AVA value than the upper-middle-income level. In 2007, low-income countries' average AVA value decreased by five per cent, then increased by four per cent.

In contrast, the average EA percentage is higher in low-income levels and descending in the values shown in Fig 2 for lower-middle, upper-middle, and high-income groups. The utilisation of FC is demonstrated to be a low average of kilograms per hectare of arable land values in low-income countries and high average FC values in high-income countries, as depicted in Fig 3. Between 2002 and 2009, the average FC value in the high-income level increased significantly. After 2009, the average FC value gradually decreased until 2021 in the high-income group.

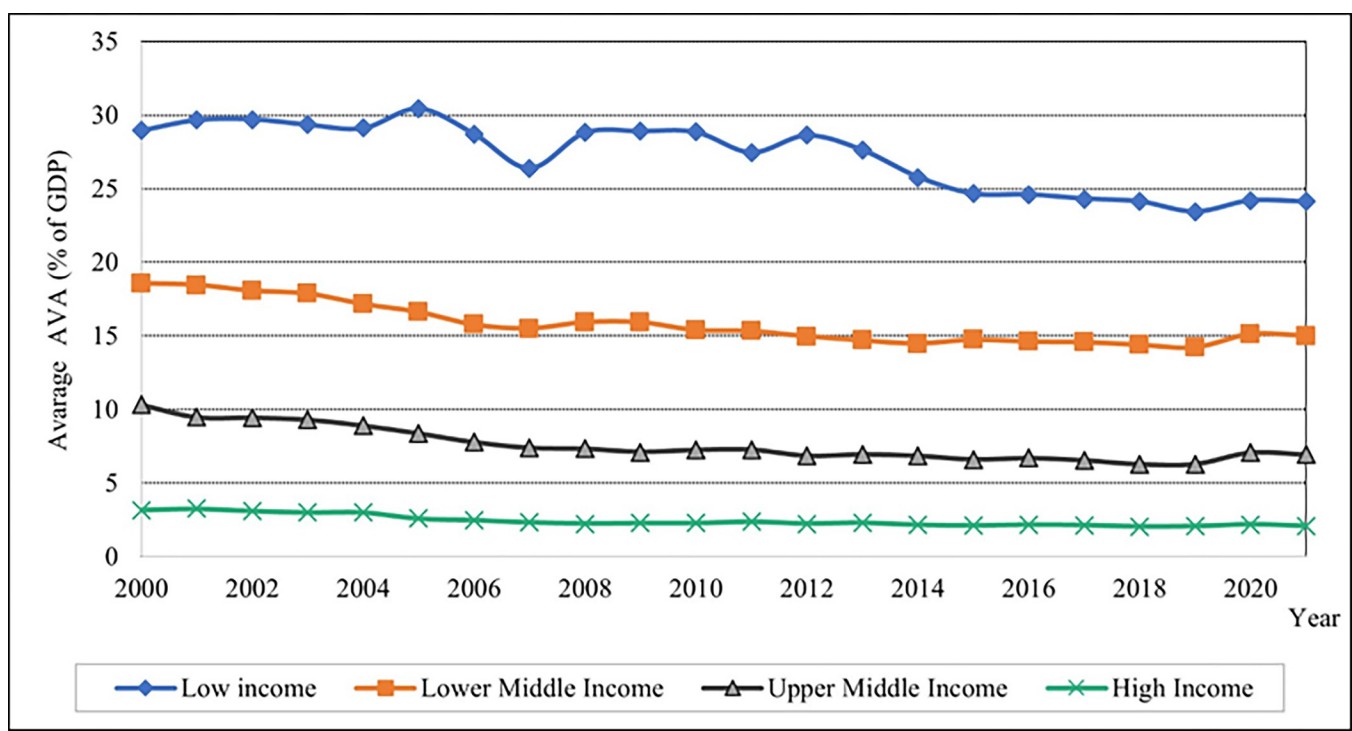

**Fig 1. Graphical depiction of agricultural value-added for each income level.** Source: Compiled by authors.

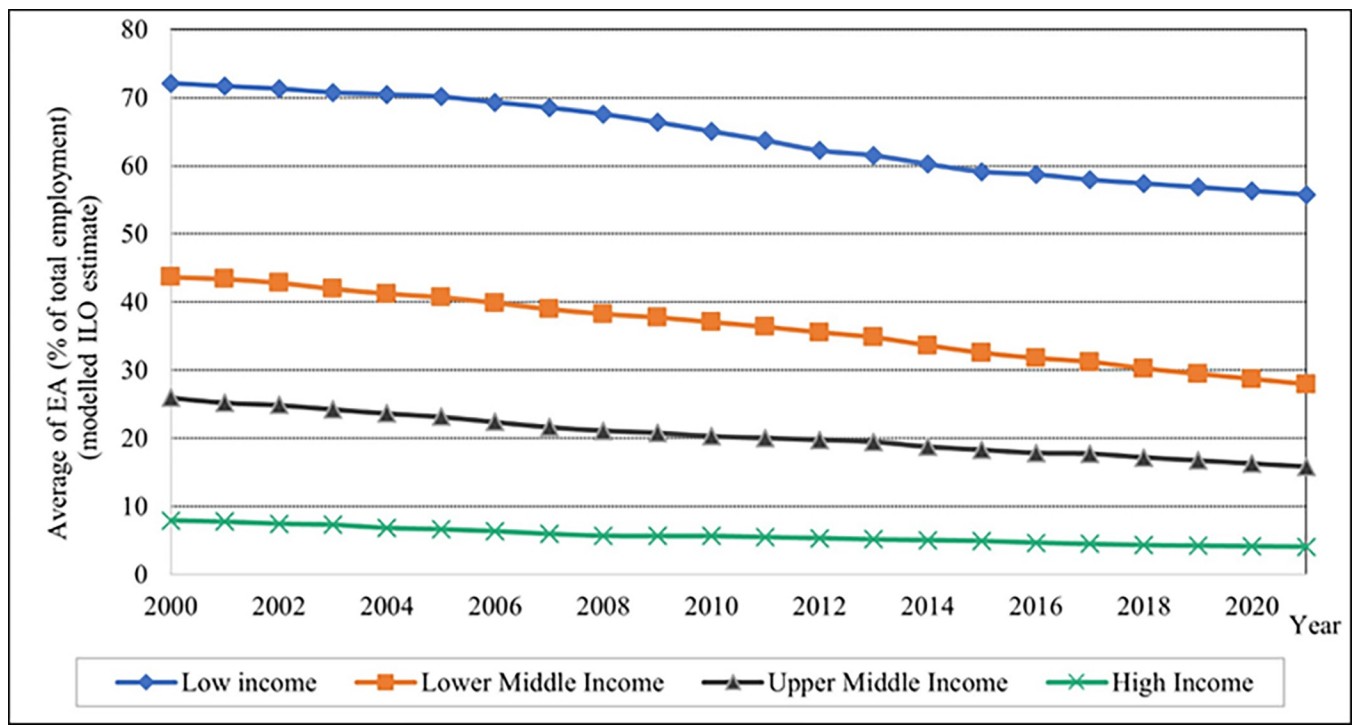

**Fig 2. Graphical depiction of employment in agriculture for each income level.** Source: Compiled by authors.

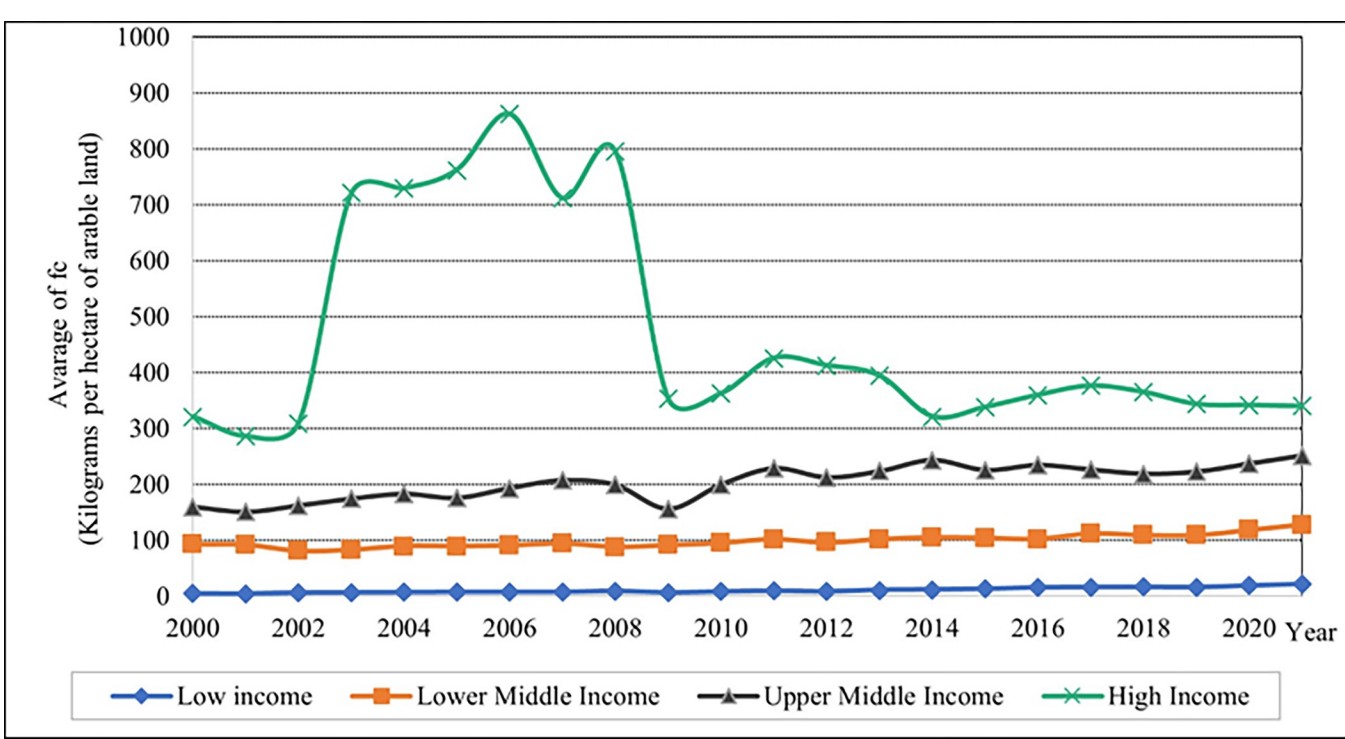

**Fig 3. Graphical depiction of fertilizer consumption for each income level.** Source: Compiled by authors.

The lower-middle-income and low-income levels have a high proportion of merchandise exports, with a varying percentage, as illustrated in Fig 4. From 2004 to 2011, the rate of average ARME values gradually decreased. However, in 2011, the percentage of average ARME value for low and lower-middle-income levels were the same.

As shown in Fig 5, the low-income levels had the highest percentage of ARMI in 2006. After 2006, the rate of ARMI fell dramatically until 2007, and the high-income group had a lower percentage of ARMI values when compared to lower-middle and upper-middle-income levels. The results indicate that high-income levels exhibit a high rate of trade, as seen in Fig 6; in ascending order, it depicts the percentage of trade values for the low, lower, and upper-middle. Compared to low-income countries, upper-middle-income countries have a high rate of GDP values.

As shown in Fig 7, the results show that lower-middle-income levels have high average ER values. After 2010, the average ER gradually increased from 2010 to 2018 and remained constant until 2021. The average ER values in high-income countries remained on the x-axis. The pattern of FDI across all income levels is shown to have varying designs, as shown in Fig 8. In 2007, the high-income group had the highest percentage of average FDI value. Compared to the upper middle-income level and high-income level percentage of average FDI exhibits a vast difference from 2004–2009, as demonstrated in this graph. Furthermore, the trend line pattern is consistent across upper and lower middle-income levels.

In conclusion, the figures presented in this study offer a profound understanding of the intricacies of the relationship between income levels and their corresponding effects on the dependent and independent variables between 2000 to 2021. The data presented in these figures serve as a valuable lens through which to observe and interpret the nuances of economic performance in various income levels, thereby providing a framework for informed decision-

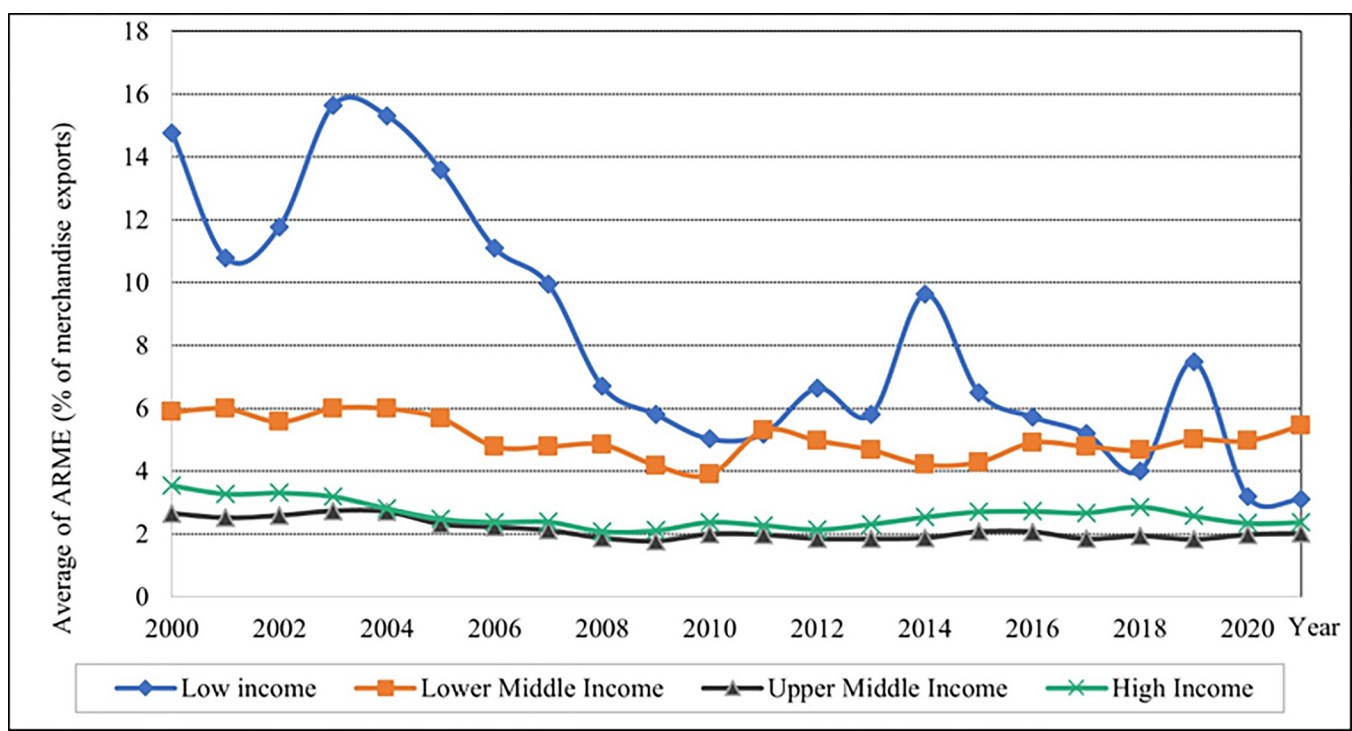

**Fig 4. Graphical depiction of agriculture raw material exports of each income level.** Source: Compiled by authors.

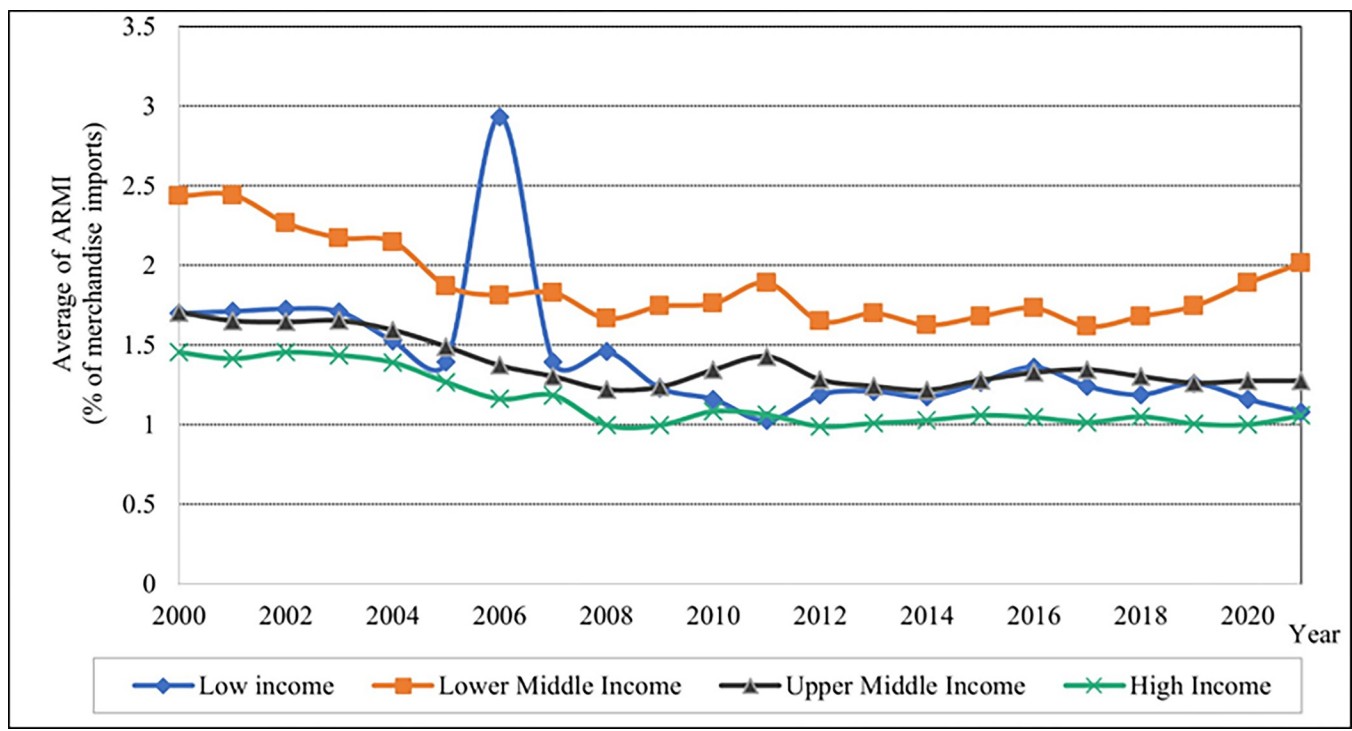

**Fig 5. Graphical depiction agriculture raw material imports of each income level.** Source: Compiled by authors.

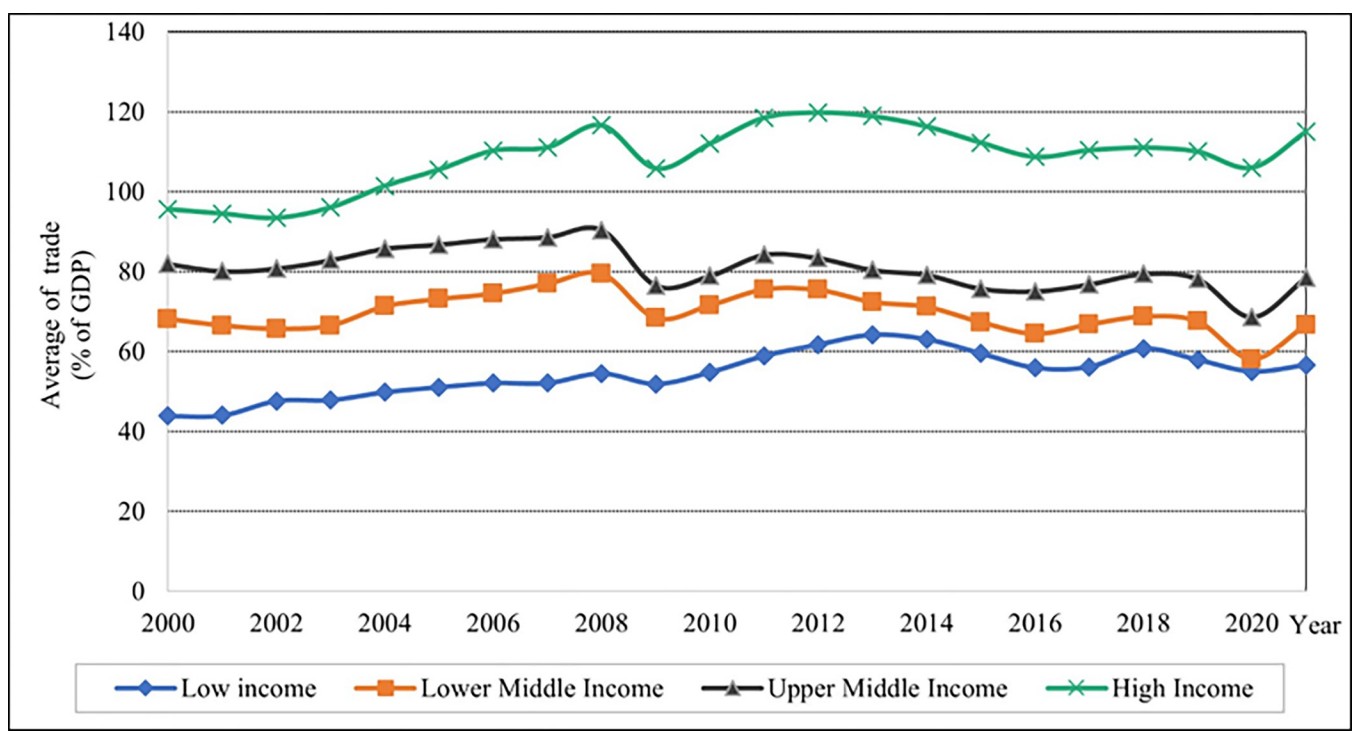

**Fig 6. Graphical depiction of trade for each income level.** Source: Compiled by authors.

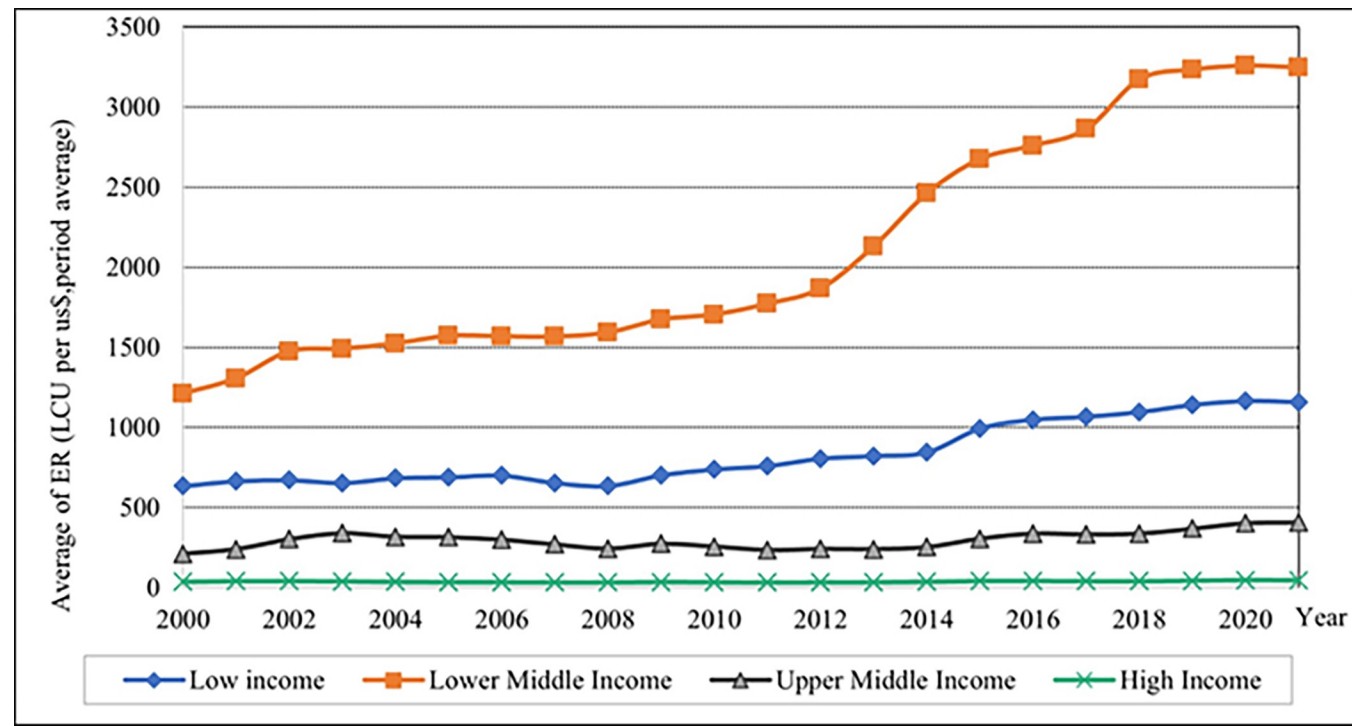

**Fig 7. Graphical depiction of exchange rate for each income level.** Source: Compiled by authors.

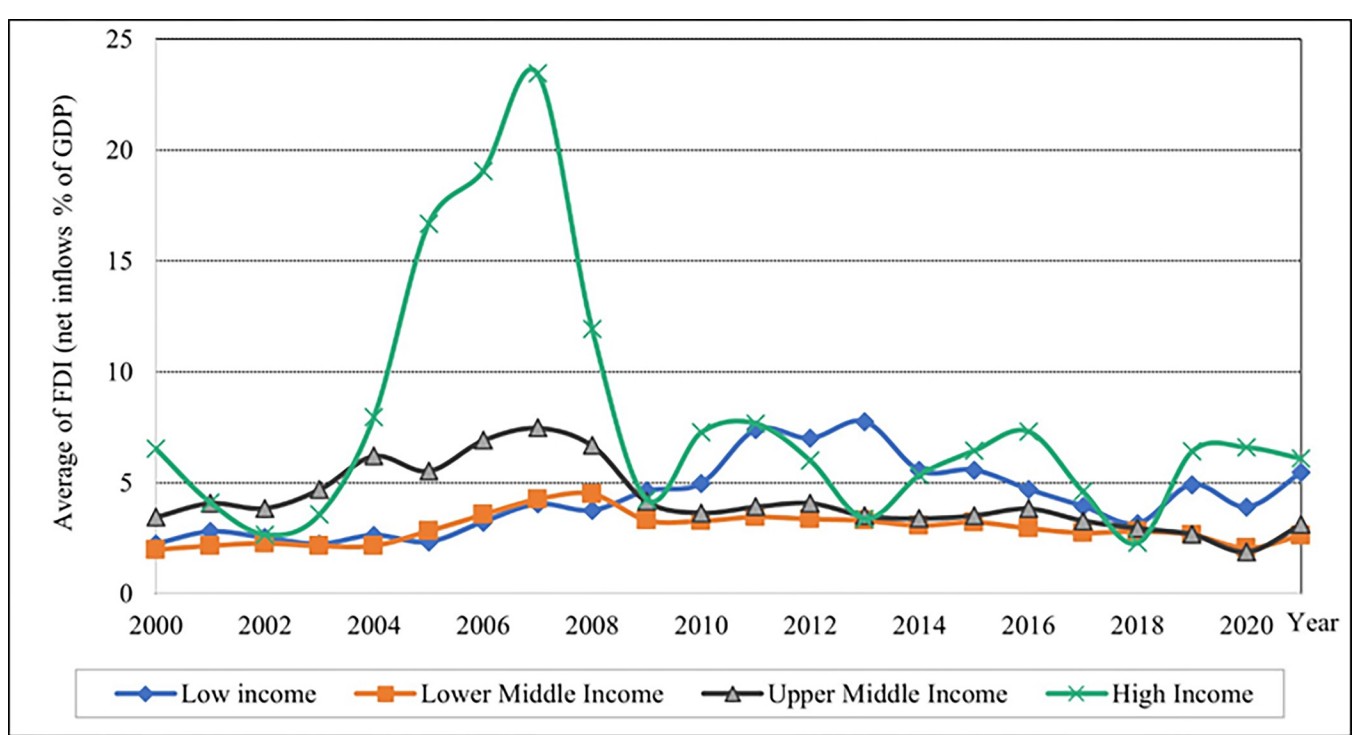

**Fig 8. Graphical depiction of foreign direct investments for each income level.** Source: Compiled by authors.

making and strategic planning in the future. By illuminating the intricate interplay between income levels and their corresponding effects, these figures shed light on the complexity of the economic landscape, thereby enabling a deeper understanding of the mechanisms at play.

Table 3 comprehensively evaluates the best model fit for panel data analysis, including all countries, high-income, low-income, lower-middle-income, and upper-middle-income countries. The results of the F test and the Breusch-Pagan LM test are used to reject the null hypothesis of the POLS model being the best analysis method. The rejection of the null hypothesis implies that the POLS model is inappropriate for this study. To determine the most suitable model, the Hausman test is conducted. The results of the Hausman test reveal that the REM outperforms the FEM for all countries, lower-middle-income and upper-middle-income levels. The rejection of the null hypotheses for these levels indicates this. On the other hand, the

**Table 3. Specification tests for final stepwise panel regression model.**

| Income Levels | Tests | | |
|---|---|---|---|
| | F test | LM Test | Hausman Test (Sigmamore) |
| | $H_0$: POLS | $H_0$: POLS | $H_0$: Random Effect |
| | $H_1$: Fixed Effect | $H_1$: Random Effect | $H_1$: Fixed Effect |
| All Countries | 1220.74*** | 14306.92*** | 31.35*** |
| High-Income | 137.96*** | 4534.82*** | 0.74 |
| Low-Income | 33.05*** | 721.21*** | 14.96*** |
| Lower-Middle Income | 165.84*** | 2830.01*** | 9.67* |
| Upper-Middle Income | 86.24*** | 2157.79*** | 9.06 |

Note: The symbols *, **and *** represents 10%, 5% and 1% significance level, respectively.

**Table 4. Selected fixed effect and random effect estimates for the final stepwise model.**

|  | All Countries | High-Income | Low-Income | Lower-Middle | Upper- Middle |
|---|---|---|---|---|---|
| Equation | Y = f (EA ARMI FC FDI ER) | Y = f (EA FDI ER ARMI FC) | Y = f (FC EA ER ARMI FDI) | Y = f (EA FDI FC ARME ER) | Y = f (EA ARME ARMI ER FDI FC) |
| Variables | AVA | AVA | AVA | AVA | AVA |
|  | FEM | REM | REM | FEM | FEM |
| FC | -6.79e-06 | -4.10e-06 | -0.0944 | 0.000841 | 0.000165 |
|  | (0.0000104) | (5.91e-06) | (0.0703) | (0.0058) | (0.00108) |
| EA | 0.2616*** | 0.2341*** | 0.1560** | 0.2692*** | 0.2523*** |
|  | (0.0476) | (0.0609) | (0.0629) | (0.0768) | (0.0469) |
| ARME |  |  |  | 0.0829 | 0.4690** |
|  |  |  |  | (0.0842) | (0.2002) |
| ARMI | 0.0029 | 0.2707 | 0.2217 |  | 0.1742 |
|  | (0.1416) | (0.1894) | (0.1646) |  | (0.3510) |
| FDI | 0.0012 | 0.000904*** | 0.0419 | -0.1536* | 0.0078 |
|  | (0.0019) | (0.000314) | (0.0713) | (0.0824) | (0.0533) |
| ER | 0.000143*** | 0.00250*** | -0.0020** | 0.000161*** | 0.00031 |
|  | (0.0000544) | (0.0007053) | (0.00085) | (0.0000464) | (0.00024) |
| Constant | 3.7150 | 0.6931 | 19.4104 | 5.6348 | 1.0143 |
| No of Countries | 101 | 32 | 11 | 28 | 30 |
| No of years | 22 | 22 | 22 | 22 | 22 |
| $R^2$ Within | 0.2865 | 0.3829 | 0.2817 | 0.3363 | 0.3976 |
| $R^2$ Between | 0.7988 | 0.5787 | 0.1888 | 0.5619 | 0.5462 |
| $R^2$ Overall | 0.7643 | 0.5500 | 0.2087 | 0.5229 | 0.4843 |

Note: The symbols *, **, and *** represents 10%, 5%, and 1% significance level, respectively. Parentheses represent the robust standard error. FE and RE represent the Fixed effect and Random effect, respectively.

REM performs better than the FEM for low-income and high-income groups, as evidenced by the failure to reject the null hypotheses for these levels.

In conclusion, Table 3 indicates that the FE model is not valid for high-income and upper-middle-income levels. In contrast, the FE model best fit for all countries and low-income groups. These findings emphasise the importance of considering the specific income level when selecting the appropriate panel data model.

Table 4 shows the final selected best models among the RE and FE model's coefficient, robust standard error values, significant groups, and $R^2$ outcomes for all countries and each country's income level. According to the results, at the global level, EA and ER have a significant impact, and ARMI, FC, and FDI have an insignificant effect on the AVA. Overall model fits that's mean $R^2$ is equal to 0.7643, which means 76.43% of the variance of the output variable (AVA) is explained by the variance of the input variables (FC, EA, ARME, ARMI, ER, FDI). The findings show that at the high-income level, EA, FDI, and ER significantly impact the AVA, whereas ARMI and FC have an insignificant impact. Its overall $R^2$ value is 0.5500. which means the model was fitted by 55%. The results demonstrate that at the low-income level, EA and ER significantly affect the AVA, while FC, ARMI, and FDI have an insignificant impact. At the lower middle-income level, EA, FDI, and ER significantly impact the AVA considering that FC and ARME have negligible effects. The difference between the $R^2$ value in the low- and lower-middle-income levels is 34.42%. Last, at the upper middle-income level, EA and ARME significantly impact the AVA, whereas ARMI, ER, FDI, and FC have an insignificant effect.

In conclusion, the findings from Table 4 show that the impact of various factors such as EA, FC, ARME, ARMI, ER, and FDI on the AVA varies depending on a country's income level. Furthermore, the impact of the factors affecting the AVA varies among different income levels, and it is essential to consider these differences when making policy decisions.

## Discussion

The findings suggest that the impact of EG factors, including FC, EA, ARMI ARME, ER, and FDI on AVA depends on a country's income level. This implies that the effect of EG factors on agriculture may differ depending on whether a country is considered high income, low-income, lower-middle-income, or upper-middle-income.

In high-income countries, the agriculture sector is often a small part of the economy but still provides food security and rural employment [66]. Moreover, the increase in FDI can provide capital for modernising and expanding the sector, leading to increased value-added. Also, Agricultural FDI has a significant promoting effect on agricultural GTFP and various sub-items. However, it has an inverted U-shaped feature in the long-term [67] and supports the present study findings, which significantly impacts FDI on AVA in the high-income level. However, the effects of Trade and ER are often limited, as the agriculture sector is less exposed to global market forces [68].

In low-income countries, the present study identifies that EA significantly impacts AVA. According to previous studies, the agriculture sector is often the population's primary source of employment and income, making it a critical sector for economic growth and poverty reduction [31,69]. The impact of globalisation factors, such as increased FC, can positively affect crop yields and agricultural production, leading to increased value-added in the sector [33] However, according to the current analysis, identifying FC is insignificant on AVA. Additionally, ER fluctuations can significantly impact the cost of inputs and the price of exports, making it difficult for farmers to plan and invest [70]. In lower middle-income countries, the agriculture sector is transforming as the economy diversifies [71]. The impact of globalisation factors, such as increased exports of agricultural raw materials, can provide new markets and increase demand, leading to increased value-added in the sector [72]. However, these countries also face significant challenges, including competition from low-cost producers, who can benefit from lower labour costs, weaker environmental regulations, and lower production costs [73]. Furthermore, the present study identifies that managing ER fluctuations is a crucial policy area for enhancing economic sustainability and growth of the agricultural sector, which supports the findings of previous studies [70,74]. ER fluctuations can significantly impact the cost of inputs and the price of exports, making it difficult for farmers to plan and invest. Additionally, increased imports of agricultural raw materials can lead to lower consumer prices and negatively impact domestic producers, who may struggle to compete with lower-priced imports.

In upper-middle-income countries, agriculture is typically less critical regarding employment and income but still plays a role in the economy [75]. The impact of globalisation factors, such as increased ARMI, can lead to increased efficiency and lower prices, benefiting consumers and increasing value-added in the sector. However, competition from low-cost producers can also pose a challenge, and the impact of ER can still be significant [76,77]. Additionally, increased ARMI can lead to lower prices for consumers and negatively impact some domestic producers, who may struggle to compete with lower-priced imports [77]. However, the present study identified ARMI as insignificant and ARME as significant on AVA. The lack of FDI in agriculture affects the sector's competitiveness in the Republic of Moldova. Both producers and the government must take action to improve the situation by focusing on product quality and attracting FDI [29].

Overall, it is essential for policymakers to consider the impact of globalisation on the agriculture sector carefully and to develop strategies to maximise the benefits while minimising the adverse effects.

## Policy implications

This study provides a holistic picture of the impact of EG on AVA, which will assist governments in the formulation, alignment and revision of their strategies and policies to expedite the growth of agro-based exports, and, in turn, the economy will have a positive impact in increasing GDP.

The findings of this study provide valuable policy implications for respective governments aiming to enhance economic development and accelerate the growth of agro-based exports. The study demonstrates that promoting EA and managing ER fluctuations are crucial policy areas at the global level. Governments should invest in creating employment opportunities in the agricultural sector, such as supporting farmers, agro-entrepreneurs, and large-scale agro-based enterprises. At the same time, policies should be implemented to manage ER fluctuations, such as reducing reliance on imports and promoting export diversification.

In high-income countries, policies should continue focusing on employment creation in the agricultural sector while promoting FDI and managing ER fluctuations. In low-income countries, the negative impact of ER fluctuations on AVA highlights the importance of policies that manage ER fluctuations to enhance the economic sustainability of smallholder farmers. Furthermore, at the lower-middle-income level, policies should focus on employment creation in the agricultural sector while promoting FDI and managing ER fluctuations. Lastly, at the upper-middle-income level, policies should promote ARME while also creating employment opportunities in the agricultural sector.

Moreover, the study reveals that FC and ARMI had no significant impact on AVA at all income levels. Therefore, governments should consider reducing dependence on fertilizer and agriculture raw material imports by developing alternative strategies to improve soil fertility and promote domestic agricultural production.

To promote sustainable agricultural development and enhance economic growth, governments must implement policies that focus on managing ER fluctuations, promoting ARME and FDI, and developing the necessary skills through vocational and tertiary education systems. This study highlights the need for policies that prioritize employment creation, managing ER fluctuations, and promote ARME to enhance the economic sustainability and growth of the agricultural sector. Investing in infrastructure and technology and fostering value chains in the industry are other important measures to consider. In addition, policies that promote vocational and tertiary education can help improve the country's productivity and profitability, thereby increasing AVA.

In conclusion, promoting sustainable agricultural development, increasing productivity and livelihoods of farmers, and creating employment opportunities are crucial for enhancing economic development. Governments must implement policies that focus on managing ER fluctuations, promoting ARME and FDI, and developing the necessary skills through vocational and tertiary education systems. These policies will promote the economic sustainability of smallholder farmers and support the growth of the agricultural sector, which is essential for the economic development of countries at all income levels.

### Future research

One potential avenue for future research is to investigate how economic factors such as EA, ER, and FDI impact AVA across countries with different income levels using with moderate

variables. By examining the effects of these factors on AVA at varying income levels, researchers can gain a better understanding of how economic policies may impact AVA in different contexts. Another potential area of focus is to explore the specific mechanisms through which EA, ER, and FDI influence AVA. This could involve analysing the various channels through which these factors impact AVA, such as through changes in labour market conditions or access to education and healthcare. When implementing their methodology, the authors can consider employing the Two Stage Least Squares (TSLS) or Generalized Method of Moments (GMM) as widely used instrumental variable (IV) estimators, in addition to the stepwise method. Overall, such research could provide valuable insights into how economic factors can impact AVA and inform policy decisions aimed at promoting greater economic and social well-being. On the other hand, could explore potential interactions and trade-offs among these factors such as trade offs between FDI and domestic investment in agriculture or between environmental sustainability and agricultural productivity. There is need to explore these country-specific mechanisms and how they interact with globalization factors to affect AVA. Moreover, in this technological era, the agricultural sector can also benefit from the use of technology to enhance productivity and value addition. Therefore, more research is needed to investigate the role of technology in the agricultural sector and how it can be leveraged to enhance AVA. Addressing these research gaps can provide valuable insights for policymakers and stakeholders to promote sustainable economic development through increased agriculture value addition. Lastly, conducting cross country comparisons and analysing differences in the impact of these factors across regions all countries with different economic, social and cultural context could provide a more comprehensive understanding of the complex relationships between these factors and AVA.

## Limitation

There are some limitations to this study that need to be acknowledged. One of these is the potential for omitted variable bias, where factors not included in the analysis could have influenced the results. To address this issue, future studies could consider a broader range of variables to account for potential confounding factors and improve the robustness of the findings. Due to data constraints, the study limited its analysis to a maximum of 101 countries and the time period between 2000 and 2021. As a result of data unavailability, observations for other countries and years could not be incorporated into the study. Furthermore, the study could benefit from including more recent data to improve the accuracy and relevance of the findings. It is also worth noting that the study's results may need to be more generalisable to other contexts, given the specific sample and methodology used.

## Conclusion

This study has several significant contributions to the field of economic globalisation and value-added agriculture. Firstly, it adds to the growing body of literature on the impact of globalisation on agriculture by providing a nuanced understanding of how different factors influence value-added agriculture across different income levels. This is especially important given agriculture's critical role in many countries' economies, particularly in developing nations.

Secondly, the study employs a novel methodology that takes into account the complex relationship between economic globalisation and value-added agriculture. Using a panel data regression with the stepwise method, the study provides a more accurate representation of the factors contributing to value-added agriculture in different income levels globally. This approach offers a better understanding of the impact of globalisation on agriculture over time, making it easier to develop effective policies that maximise the benefits of globalisation while minimising its negative impacts.

Thirdly, the study highlights the importance of considering countries' income levels when examining the impact of economic globalisation on agriculture. The findings suggest that different income levels have different drivers that impact value-added agriculture. This knowledge is crucial for policymakers to design policies tailored to different countries' specific needs.

Finally, this study contributes to a better understanding of the role of value-added agriculture in economic development. By demonstrating that value-added agriculture can significantly impact income levels, the study provides valuable insights for policymakers aiming to promote economic growth and development in their countries.

In conclusion, this study contributes to the literature on economic globalisation and value-added agriculture. It offers insights into the complex relationship between globalisation and agriculture. It highlights the importance of considering different income levels when designing policies to maximise the benefits of globalisation while minimising its negative impacts. This study's findings have important implications for policymakers and other stakeholders, making it a valuable resource for future research.

## Supporting information

**S1 Appendix. Data file.**
(XLSX)

**S2 Appendix. Specification test results for global and different income groups.**
(DOCX)

**S3 Appendix. Results of panel regression for global and different income groups.**
(DOCX)

**S4 Appendix. Fixed effect and random effect estimates for the final stepwise model.**
(DOCX)

## Author Contributions

**Conceptualization:** Nadeena Sansika, Raveesha Sandumini, Chamathka Kariyawasam, Tharushi Bandara, Krishantha Wisenthige, Ruwan Jayathilaka.

**Data curation:** Nadeena Sansika, Raveesha Sandumini, Chamathka Kariyawasam, Tharushi Bandara, Ruwan Jayathilaka.

**Formal analysis:** Nadeena Sansika, Raveesha Sandumini, Chamathka Kariyawasam, Tharushi Bandara, Krishantha Wisenthige, Ruwan Jayathilaka.

**Investigation:** Nadeena Sansika, Raveesha Sandumini, Chamathka Kariyawasam, Tharushi Bandara, Ruwan Jayathilaka.

**Methodology:** Raveesha Sandumini, Chamathka Kariyawasam, Ruwan Jayathilaka.

**Project administration:** Krishantha Wisenthige, Ruwan Jayathilaka.

**Resources:** Ruwan Jayathilaka.

**Supervision:** Krishantha Wisenthige, Ruwan Jayathilaka.

**Validation:** Krishantha Wisenthige, Ruwan Jayathilaka.

**Writing – original draft:** Nadeena Sansika, Raveesha Sandumini, Chamathka Kariyawasam, Tharushi Bandara, Krishantha Wisenthige, Ruwan Jayathilaka.

**Writing – review & editing:** Krishantha Wisenthige, Ruwan Jayathilaka.

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
