## [Decision Letter · Decision Letter 0]

22 Mar 2023

PONE-D-23-05826Impact of Economic Globalisation on Value-Added Agriculture, GloballyPLOS ONE

Dear Dr. Jayathilaka,

Thank you for submitting your manuscript to PLOS ONE. After careful consideration, we feel that it has merit but does not fully meet PLOS ONE’s publication criteria as it currently stands. Therefore, we invite you to submit a revised version of the manuscript that addresses the points raised during the review process. Please submit your revised manuscript by May 06 2023 11:59PM. If you will need more time than this to complete your revisions, please reply to this message or contact the journal office at plosone@plos.org. Please include the following items when submitting your revised manuscript:A rebuttal letter that responds to each point raised by the academic editor and reviewer(s). You should upload this letter as a separate file labeled 'Response to Reviewers'.A marked-up copy of your manuscript that highlights changes made to the original version. You should upload this as a separate file labeled 'Revised Manuscript with Track Changes'.An unmarked version of your revised paper without tracked changes. You should upload this as a separate file labeled 'Manuscript'.

We look forward to receiving your revised manuscript.

Kind regards,

Hoang Phong Le, Ph.D.

Academic Editor

PLOS ONE

Journal Requirements:

Additional Editor Comments:

Five reviewers have commented on your paper. While they see its potential, they point out a number of remarks and weaknesses that need to be addressed carefully. In particular, I stress your attention on the following points:

Emphasize the novelty and impactful contribution of this work as currently this appears to be marginal.Review the newly published papers.Make clear about the theoretical underpinning.Explain more the reason for the selection of sample. You need to better explain your variable. How you used available data to proxy your variable.Add research questions; formalize your hypotheses and link your results to them.Highlight your improvement of the method and your innovation in methods. Endogeneity problem or cross-sectional dependence should be taken into account in the study.Improve discussions, policy implications, and conclusion sections.Revise and improve the English language used in the paper.

Please include a point-by-point reply to the above comments, alongside the reply to the reviewers' comments.

Reviewers' comments:

Reviewer's Responses to Questions

**Comments to the Author**

1. Is the manuscript technically sound, and do the data support the conclusions?

Reviewer #1: Partly

Reviewer #2: Yes

Reviewer #3: No

Reviewer #4: No

Reviewer #5: Partly

2. Has the statistical analysis been performed appropriately and rigorously? 

Reviewer #1: No

Reviewer #2: Yes

Reviewer #3: No

Reviewer #4: Yes

Reviewer #5: No

3. Have the authors made all data underlying the findings in their manuscript fully available?

Reviewer #1: Yes

Reviewer #2: Yes

Reviewer #3: No

Reviewer #4: Yes

Reviewer #5: Yes

4. Is the manuscript presented in an intelligible fashion and written in standard English?

Reviewer #1: No

Reviewer #2: Yes

Reviewer #3: No

Reviewer #4: No

Reviewer #5: Yes

5. Review Comments to the Author

Reviewer #1: The author has presented a well-written article (Impact of Economic Globalisation on Value-Added Agriculture, Globally), detailed, and supported by solid analysis. However, reviewers must make some suggestions to improve this manuscript's quality, including:

1. The manuscript does not have a strong theoretical foundation. I recommend the author add international trade theory and how the research variable represents the theory.

2. The author uses 101 countries and categorized them based on GDP per capita, but the authors need to mention the source of the categorization: What is the World Bank, IMF or other sources?

3. Why do the authors make different models for each country category? This makes the conclusion and policy implications inappropriate. Moreover, the assumption of economic globalization is all countries face the same external conditions so the author needs to make the same model between countries. Literature review and incomplete data cannot be the justification for differences in models between countries.

4. The author is expected to check again Table 3. For example, is it right to choose RE for all countries model? Even though the F-Test and Hausman Test show H0 is rejected so the best model is Fixed Effect.

5. The author should also choose only 1 best model (RE or FE) to be displayed in Table 4. In addition, what is meant by numbers in brackets in Table 4? probability value or standard error? If it is the probability: several variables should be significant (p-value is smaller than 0.05).

6. Please the authors to improve discussions, policy implications, and conclusion sections based on the revision of the results section

7. The author needs to add limitations and further research in the conclusion section.

8. Writing references must be fixed to adjust the PLOS ONE journal template

9. The author should use a professional language editor to improve the quality of this manuscript.

Finally, I hope these various suggestions can help you improve this manuscript's quality and can be published in the Plos One journal. Good luck!

Reviewer #2: The panel data regression with the stepwise method was employed to quantify the impact of economic globalisation in 101 countries between 2000 to 2021. The results show that agricultural employment significantly impacts the agricultural value-added factor globally and across all income levels. Also, countries with low and lower-middleincome levels significantly affect agricultural value-added due to exchange rates. In comparison, high-income and lower-middle-income levels have an impact due to foreign direct investment. Finally, the upper-middle-income countries have significantly affected agricultural value-added due to agricultural raw materials imports. The study is well presented, i have some comments on it, i.e.,

1) Title f the study should be changed, for instance, "The Role of Economic Globalization in the Transformation of Agricultural Value Chains"

2) Introduction: Add possible research questions and linked them with the study's objectives.

3) Add latest literature up to 2023, for instance,

- Gyamfi, B. A., Onifade, S. T., Erdoğan, S., & Ali, E. B. (2023). Colligating ecological footprint and economic globalization after COP21: Insights from agricultural value-added and natural resources rents in the E7 economies. International Journal of Sustainable Development & World Ecology, 1-15.

- Zaman, K. (2023). The Future of Financial Support for Developing Countries: Regional and Islamic Monetary Funds. Politica, 1(1), 1–8. https://doi.org/10.5281/zenodo.7610145

- Raihan, A. (2023). An econometric evaluation of the effects of economic growth, energy use, and agricultural value added on carbon dioxide emissions in Vietnam. Asia-Pacific Journal of Regional Science, 1-32.

- Khan, M. (2023). Shifting Gender Roles in Society and the Workplace: Implications for Environmental Sustainability. Politica, 1(1), 9–25. https://doi.org/10.5281/zenodo.7634130

- Chen, Y., & Zhang, Y. (2023). Services Development, Technological Innovation, and the Embedded Location of the Agricultural Global Value Chain. Sustainability, 15(3), 2673.

- Fatima, S. (2023). Rural Development and Education: Critical Strategies for Ending Child Marriages. Archives of the Social Sciences: A Journal of Collaborative Memory , 1(1), 1-15. https://doi.org/10.5281/zenodo.7556588

- Xu, Y., Li, C., & Wang, J. How does agricultural global value chain affect ecological footprint? The moderating role of environmental regulation. Sustainable Development.

- Aqib, M., & Zaman, K. (2023). Greening the Workforce: The Power of Investing in Human Capital. Archives of the Social Sciences: A Journal of Collaborative Memory, 1(1), 31–51. https://doi.org/10.5281/zenodo.7620041

- Raihan, A., Muhtasim, D. A., Farhana, S., Hasan, M. A. U., Pavel, M. I., Faruk, O., ... & Mahmood, A. (2023). An econometric analysis of Greenhouse gas emissions from different agricultural factors in Bangladesh. Energy Nexus, 100179.

- Khan, M, T., & Imran, M. (2023). Unveiling the Carbon Footprint of Europe and Central Asia: Insights into the Impact of Key Factors on CO2 Emissions. Archives of the Social Sciences: A Journal of Collaborative Memory, 1(1), 52–66. https://doi.org/10.5281/zenodo.7669782

- Yan, B., Xia, Y., & Jiang, X. (2023). Carbon Productivity and Value-Added Generations: Regional Heterogeneity along Global Value Chain?. Structural Change and Economic Dynamics.

4) Literature review: Add missing gaps and contribution of the study.

5) Add possible research hypotheses in the literature review.

6) Compare and contrast the used statistical panel technique with the cross-panel techniques, for instance,

- Guschanski, A., & Onaran, Ö. (2023). Global Value Chain Participation and the Labour Share: Industry‐level Evidence from Emerging Economies. Development and Change, 54(1), 31-63.

- Zaman, K. (2023). A Note on Cross-Panel Data Techniques. Latest Developments in Econometrics, 1(1), 1-7. https://doi.org/10.5281/zenodo.7565625

- Zhang, Y., Liao, C., & Pan, B. (2023). Ecological unequal exchange between China and European Union: An investigation from global value chains and carbon emissions viewpoint. Atmospheric Pollution Research, 101661.

7) Add possible research limitations and future directions at the end.

Reviewer #3: Review report PONE-D-23-05826 Impact of Economic Globalisation on Value-Added Agriculture, Globally

Abstract

The topic of this article is Impact of Economic Globalisation on Value-Added Agriculture. However, the abstract does not contain any information about economic globalisation.

“The panel data regression with the stepwise method was employed to quantify the impact of economic globalisation in 101 countries between 2000 to 2021.” Quantify the impact of economic globalisation on? Grammatical error.

Introduction

“In addition, Globalisation can transform rural agriculture into more commercialised and value-based agriculture and improve the rural community’s living conditions [5].” Why Globalisation is a capital letter?

Are these variables refer to economic globalisation? “Trade, Exchange Rate (ER) and Foreign Direct Investment (FDI) are crucial in the global agricultural sector.” If yes, please write it and inform readers.

The in-text citation is not consistent throughout the article. Example, “Nyiwul and Koirala [7].”, “Manamba Epaphra (2017)”, “Schuh [10]”

This study is about global. Why specifically mentioned about China? “In China, according to a performance index, agriculture is receiving more foreign FDI, but not at a rate that is satisfactory given the size of the industry [8, 9].”?

Why did you mention this? “Further findings demonstrate that Employment in Agriculture (EA) and Fertilizer Consumption (FC) affect AVA.”

Any intention to include this in the article? “Agriculture Raw Material Exports (ARME) and Agriculture Raw Material Imports (ARMI) effects the foreign currency inflows and outflows of nations.”

This section is poorly written. The authors introduced all the keywords, then proceed to write the purpose of the study. This research is not supported by strong research problem.

Literature Review

Since the authors did not guide the readers to understand the research problem well from the beginning, the contents of this section are also unable to add value to the readers.

It is very obvious that the authors only summarise the past literature on the variables that the authors intend to study. The authors were unable to provide a critical evaluation of these works.

Why break the section into High-income, Low-Income Level, Lower Middle-Income Level and Upper Middle-Income Level?

This section is poorly written. The authors failed to integrate arguments into the review. Authors simply summarising their readings.

Data and Methodology

This section is weakly written. The authors only present the variables intended to study. There is no theoretical framework or hypothesis to support the methods. All the variables and methods are basically formed by the readers without any literature support.

Results

This manuscript only stated the findings without interpretation.

Discussion

This section is poorly written. Too much description and not enough analysis. The authors basically cited a lot of literature to end a sentence.

Policy Implications

This section is incompetently written. The authors basically mentioned “the government should …” for all the variables examined in this study.

Conclusion

The authors only summarise the study.

The conclusion is intended to help the reader understand why your research should matter to them after they have finished reading the paper. A conclusion is not merely a summary of your points or a re-statement of your research problem but a synthesis of key points.

Overall

It is a poorly written article. It seems like an assignment for undergraduate students. It does not meet the expectation to be published in a high-impact journal.

Reviewer #4: This manuscript reports on (PONE-D-23-05826) “Impact of Economic Globalisation on Value-Added Agriculture, Globally”. I want to suggest a few suggestions to improve the manuscript's quality and better readability.

(I) The English language needs more work. There are many grammatical and typo mistakes in this manuscript. The paper needs to be edited by a native English speaker.

(II) I noticed that the novelty of this paper is not described in detail. This should be put in the introduction section properly. There is a need to do a more rigorous an asymmetric literature review. It should update literature to current... The authors should clearly mention the literature gap.

(III) I would like to suggest that authors should update the introduction and results part. Specifically, the latest research trends, and in order to highlight the academic frontier of the research, the references of the recent year need to be referenced.

https://doi.org/10.1016/j.renene.2021.07.014, https://doi.org/10.1002/pa.2712, https://doi.org/10.3390/su12072930, https://doi.org/10.1016/j.energy.2021.122515, https://doi.org/10.1016/j.renene.2021.10.067, https://doi.org/10.1016/j.pnucene.2022.104533, https://doi.org/10.1007/s13762-022-04638-2, https://doi.org/10.1007/s11356-022-23179-2.

(IV) How did the authors get from the theoretical model to the empirical one? Behind the model there need to be a complete and well-thought-out theoretical grounding part of the article shouldn't include any citations or references; rather, it should structured according to the authors' reasoning. The empirical model will come when this part has been completed.

(V) The authors should present the main findings in graphical form. It will increase the brevity and more readerships and attract more audience.

Reviewer #5: Review comments for the paper entitled “Impact of Economic Globalisation on Value-Added Agriculture, Globally”.

I find the research paper considers a very important study area and well written. The presentation of the work is very good and the research topic is very important from the current development perspective.

However, I have concerned for the following points:

1. It is not clear why study period is chosen from 2000 to 2021.

2. Model selection is not correct as there could be endogenity problem. Therefore, GMM method is essential. FE and RE models cannot taken seriously in this case.

3. Choice of variables also not clear. It has to clearly mention for different paragraph for different variables.

4. Introduction part is not clear why study area is important for the research. Need to rewrite with mentioning the importance and research questions and application of the work.

5. References in the text; mix of number and title of the authors. Should be rewritten as per the journal style.

6. PLOS authors have the option to publish the peer review history of their article (what does this mean?). If published, this will include your full peer review and any attached files.

Reviewer #1: **Yes: **Agus Dwi Nugroho

Reviewer #2: No

Reviewer #3: No

Reviewer #4: No

Reviewer #5: No

---

## [Author Response · Author response to Decision Letter 0]

5 May 2023

Point by point response to reviewers

Dear Editor and Reviewers,

We would like to express our profound appreciation to the reviewers for the valuable comments and suggestions made on our manuscript which were very helpful in revising and improving it. Please note that the line numbers referred in this document is aligned with the revised manuscript which has track changes.

Reviewer 1 comment 1: The manuscript does not have a strong theoretical foundation. I recommend the author add international trade theory and how the research variable represents the theory.

Authors’ response to reviewer 1 comment 1: Thanks for this. Having taken due note of your feedback, we have now included two well-established trade theories, the Heckscher-Ohlin theory and the New Trade theory. We appreciate your valuable feedback which now strengthens the theoretical foundation of the research article. We have also explained how the research variables represent these theories. This update can be found in the revised manuscript on lines 72 to 82.

“…There are several interrelated trade theories that impact AVA. Two of the most influential theories are the Heckscher-Ohlin theory and the New Trade theory. The Heckscher-Ohlin theory suggests that countries will specialize in producing goods that use their abundant factors of production, while the New Trade theory suggests that economies of scale and increasing returns to specialisation can create a comparative advantage and lead to trade. Both theories can lead to increased value addition in agriculture as countries and firms specialise in producing goods where they have a comparative advantage and invest in research and development to improve the quality of their products. In summary, international trade can increase value addition in agriculture…”

Reviewer 1 comment 2: The author uses 101 countries and categorized them based on GDP per capita, but the authors need to mention the source of the categorization: What is the World Bank, IMF or other sources?

Authors’ response to reviewer 1 comment 2: Thanks for the comment. Authors have used four income- level categorizations of countries by the World Bank based on their Gross National Income (GNI) per capita. 

1. Low-income countries: with a GNI per capita of $1,045 or less.

2. Lower-middle-income countries: with a GNI per capita between $1,046 and $4,125.

3. Upper-middle-income countries: with a GNI per capita between $4,126 and $12,735.

4. High-income countries: with a GNI per capita of $12,736 or more.

Comment has been incorporated in the revised manuscript and base categorization mentioned from lines 406 to 411.

Reviewer 1 comment 3: Why do the authors make different models for each country category? This makes the conclusion and policy implications inappropriate. Moreover, the assumption of economic globalization is all countries face the same external conditions, so the author needs to make the same model between countries. Literature review and incomplete data cannot be the justification for differences in models between countries.

Authors’ response to reviewer 1 comment 3: Thanks for the valuable comment made. Authors have used all variables in the initial model at each income level. However, to finalise the final model of each income level, the study utilised the stepwise method. This enables managing large amount of potential predictor variables, fine- tuning the model to choose the best predictor variables from the available options in each income category. The Stepwise technique can simplify the model, reduce overfitting, improve prediction accuracy and avoid irrelevant or redundant variables that do not contribute to the explanation of the Agriculture Value Added. Hence, the authors believed that best final models generated by the stepwise technique would be most appropriate to compare each income category. 

Reviewer 1 comment 4: The author is expected to check again Table 3. For example, is it right to choose RE for all countries model? Even though the F-Test and Hausman Test show H0 is rejected so the best model is Fixed Effect.

Authors’ response to reviewer 1 comment 4: Thanks very much. This was corrected in revised manuscript in line 651 to 655. 

“In conclusion, Table 3 indicates that the FE model is not valid for -income and upper-middle-income levels. In contrast, the FE model best fit for all countries and low-income groups. These findings emphasise the importance of considering the specific income level when selecting the appropriate panel data model”.

Reviewer 1 comment 5: The author should also choose only 1 best model (RE or FE) to be displayed in Table 4. In addition, what is meant by numbers in brackets in Table 4? probability value or standard error? If it is the probability: several variables should be significant (p-value is smaller than 0.05).

Authors’ response to reviewer 1 comment 5: Well noted. Table 4 was moved to appendix S4 and only 1 best model (RE or FE) for global level and income levels are shown in Table 4 in revised manuscript. Responding to your query, the numbers within the brackets “()” in Table 4 represent the robust standard errors and it is mentioned under the table.

Reviewer 1 comment 6: Please the authors to improve discussions, policy implications, and conclusion sections based on the revision of the results section

Authors’ response to reviewer 1 comment 6: Thanks for the valuable feedback. We have thoroughly revised the discussion, policy implications, and conclusion sections, based on the revised results section from line number 681 to 930. We consider that our revised sections provide a more insightful and comprehensive understanding of the significance of our study.

Reviewer 1 comment 7: The author needs to add limitations and further research in the conclusion section.

Authors’ response to reviewer 1 comment 7: Thank you for the feedback. Do refer to lines 843 to 835 for a detailed discussion on the limitations of the study and suggestions for future research. We believe these additions will enhance the overall contribution to our research.

Reviewer 1 comment 8: Writing references must be fixed to adjust the PLOS ONE journal template

Authors’ response to reviewer 1 comment 8: We have made the changes accordingly as per the PLOS One journal template.

Reviewer 1 comment 9: The author should use a professional language editor to improve the quality of this manuscript.

Authors’ response to reviewer 1 comment 9: Noted with thanks. The paper has been revised thoroughly and in-depth copy edit has conducted by an experienced copy-editor. We trust that the revised manuscript is free from any language errors. 

Reviewer 2 comment 1: Title of the study should be changed, for instance, "The Role of Economic Globalization in the Transformation of Agricultural Value Chains"

Authors’ response to reviewer 2 comment 1: Thanks very much for the suggested title. The purpose of current study is to investigate whether economic globalisation has had an impact on agriculture value added globally. Correspondingly, the study has less focus on agriculture value chains. Hence, the authors believed that current title is most appropriate

Reviewer 2 comment 2: Introduction: Add possible research questions and linked them with the study's objectives.

Authors’ response to reviewer 2 comment 2: Thank you for your comment. We have taken cognisance and have included possible research questions that are linked to the study's objectives. Do refer to the revised manuscript from lines 162 to 174. We trust that such additions have addressed your concern and improved the overall quality of the research.

Reviewer 2 comment 3: Add latest literature up to 2023, for instance,

- Gyamfi, B. A., Onifade, S. T., Erdoğan, S., & Ali, E. B. (2023). Colligating ecological footprint and economic globalization after COP21: Insights from agricultural value-added and natural resources rents in the E7 economies. International Journal of Sustainable Development & World Ecology, 1-15.

- Zaman, K. (2023). The Future of Financial Support for Developing Countries: Regional and Islamic Monetary Funds. Politica, 1(1), 1–8. https://doi.org/10.5281/zenodo.7610145

- Raihan, A. (2023). An econometric evaluation of the effects of economic growth, energy use, and agricultural value added on carbon dioxide emissions in Vietnam. Asia-Pacific Journal of Regional Science, 1-32.

- Khan, M. (2023). Shifting Gender Roles in Society and the Workplace: Implications for Environmental Sustainability. Politica, 1(1), 9–25. https://doi.org/10.5281/zenodo.7634130

- Chen, Y., & Zhang, Y. (2023). Services Development, Technological Innovation, and the Embedded Location of the Agricultural Global Value Chain. Sustainability, 15(3), 2673.

- Fatima, S. (2023). Rural Development and Education: Critical Strategies for Ending Child Marriages. Archives of the Social Sciences: A Journal of Collaborative Memory , 1(1), 1-15. https://doi.org/10.5281/zenodo.7556588

- Xu, Y., Li, C., & Wang, J. How does agricultural global value chain affect ecological footprint? The moderating role of environmental regulation. Sustainable Development.

- Aqib, M., & Zaman, K. (2023). Greening the Workforce: The Power of Investing in Human Capital. Archives of the Social Sciences: A Journal of Collaborative Memory, 1(1), 31–51. https://doi.org/10.5281/zenodo.7620041

- Raihan, A., Muhtasim, D. A., Farhana, S., Hasan, M. A. U., Pavel, M. I., Faruk, O., ... & Mahmood, A. (2023). An econometric analysis of Greenhouse gas emissions from different agricultural factors in Bangladesh. Energy Nexus, 100179.

- Khan, M, T., & Imran, M. (2023). Unveiling the Carbon Footprint of Europe and Central Asia: Insights into the Impact of Key Factors on CO2 Emissions. Archives of the Social Sciences: A Journal of Collaborative Memory, 1(1), 52–66. https://doi.org/10.5281/zenodo.7669782

- Yan, B., Xia, Y., & Jiang, X. (2023). Carbon Productivity and Value-Added Generations: Regional Heterogeneity along Global Value Chain?. Structural Change and Economic Dynamics.

Authors’ response to reviewer 2 comment 3: Thank you for suggesting these papers. We have duly incorporated several of them into our Introduction and Literature Review sections.

“…Another study explores the relationship between economic globalization, agricultural value-added, and ecological footprint in the E7 nations which are Brazil, China, India, Indonesia, Mexico, Russia and Turkey. The results indicate that these factors have contributed to environmental deterioration, and policymakers should implement environmental damage cost and maintain strategic resource control measures for a sustainable environment”. https://doi.org/10.1080/13504509.2023.2166141

This was corrected in revised manuscript in line 206 to 211.

“…. Another study by Raihan (2023) suggested that Vietnam should enhance AVA by introducing modern agro-based technology and promoting sustainable agriculture, such as low-carbon agriculture systems and the use of renewable energy, and by avoiding excessive use of fertilizers and pesticides.”. https://doi.org/10.5281/zenodo.7634130

This was corrected in revised manuscript in line 349 to 352.

“…Agriculture is vital for a country's economy, and enhancing the embedded location of agricultural value chains is crucial for its modernization. The global division of labor has led to an increase in the role of the agricultural global value chain, shifting agricultural trade from single-country production to multi-country production. This has refined the international division of labor in agriculture and extended production chains, leading to national agriculture being included in the global division of labor system dominated by multinational corporations.” https://doi.org/10.5281/zenodo.7556588

This was corrected in revised manuscript in line 113 to 120.

“…Another study by Raihan [57] suggested that Vietnam should enhance AVA by introducing modern agro-based technology and promoting sustainable agriculture, such as low-carbon agriculture systems and the use of renewable energy, and by avoiding excessive use of fertilizers and pesticides.”

https://doi.org/10.5281/zenodo.7620041

This was corrected in revised manuscript in line 349 to 352.

“Prior investigations found a positive association between FC and GHG emissions, indicating a need to avoid excessive use of fertilizers and pesticides in sustainable agriculture. The government should impose restrictions on the use of chemical fertilizers and engage in research and development to develop environmentally sustainable fertilizers and new crops that do not rely on hazardous fertilizers. Organic and low-carbon agriculture systems should be encouraged to reduce emissions and improve carbon sequestration. The study suggests policymakers promote organic farming, tunnel farming, no-till farming, and limit fertilizer use to reduce environmental impact.” https://doi.org/10.1016/j.nexus.2023.100179

This was corrected in revised manuscript in line 120 to 127.

Reviewer 2 comment 4: Literature review: Add missing gaps and contribution of the study.

Authors’ response to reviewer 2 comment 4: Your comment is welcome, and thus we have strengthened the contribution of the study and literature gap in the revised manuscript from lines 142 to 187. 

Reviewer 2 comment 5: Add possible research hypotheses in the literature review.

Authors’ response to reviewer 2 comment 5: Thank you for your suggestion to include possible research hypotheses in our literature review. Our study investigates the impact of economic globalization on agriculture value added at four income levels (low, lower middle, upper middle, and high), as well as at a global level. Accordingly, we have tested the significance of seven independent variables, resulting in a total of 35 hypotheses. While we appreciate the importance of including such hypotheses in a research paper, we were concerned that listing all 35 could disrupt the flow of the literature review. We have provided clear research objectives and questions and believe that our readers will understand the implied hypotheses. This has been amended in line 392 to 397.

Reviewer 2 comment 6: Compare and contrast the used statistical panel technique with the cross-panel techniques, for instance,

- Guschanski, A., & Onaran, Ö. (2023). Global Value Chain Participation and the Labour Share: Industry‐level Evidence from Emerging Economies. Development and Change, 54(1), 31-63.

- Zaman, K. (2023). A Note on Cross-Panel Data Techniques. Latest Developments in Econometrics, 1(1), 1-7. https://doi.org/10.5281/zenodo.7565625

- Zhang, Y., Liao, C., & Pan, B. (2023). Ecological unequal exchange between China and European Union: An investigation from global value chains and carbon emissions viewpoint. Atmospheric Pollution Research, 101661.

Authors’ response to reviewer 2 comment 6: Thank you for your feedback. We have revised our manuscript by including a comparison of the statistical panel technique with cross-panel techniques and citing relevant articles to support our discussion. 

“Panel data regression combines cross-sectional and time-series data to analyse changes in variables within and between entities over time. Cross-panel analysis is a type of panel data analysis that compares variable interdependence across different nations, allowing researchers to evaluate policy effects and identify relationships between variables across various periods and cross-sections. Cross-panel regression provides valuable insights into the global impact of policies by exploring the simultaneous movement of both dependent and independent variables. The utilization of the cross-panel technique is also viable for conducting research on this area” https://doi.org/10.5281/zenodo.7565625

This was added in revised manuscript in line 384 to 391.

Reviewer 2 comment 7: Add possible research limitations and future directions at the end.

Authors’ response to reviewer 2 comment 7: Duly Noted, we have added limitations and further research in before the conclusion section. You can refer from line 843 to 877.

Reviewer 3 comment 1: Abstract

The topic of this article is Impact of Economic Globalisation on Value-Added Agriculture. However, the abstract does not contain any information about economic globalisation.

“The panel data regression with the stepwise method was employed to quantify the impact of economic globalisation in 101 countries between 2000 to 2021.” Quantify the impact of economic globalisation on? Grammatical error.

Authors’ response to reviewer 3 comment 1: Thank you for highlighting this. We regret the oversight in the abstract and accordingly have made the necessary revisions to clearly underline the meaning of economic globalization. Do refer to the updated abstract for further details in lines 28 to 30. Additionally, we have corrected the grammatical error in the sentence you have mentioned, refer line to 33 to 37.

Reviewer 3 comment 2: Introduction

“In addition, Globalisation can transform rural agriculture into more commercialised and value-based agriculture and improve the rural community’s living conditions [5].” Why Globalisation is a capital letter?

Are these variables refer to economic globalisation? “Trade, Exchange Rate (ER) and Foreign Direct Investment (FDI) are crucial in the global agricultural sector.” If yes, please write it and inform readers.

The in-text citation is not consistent throughout the article. Example, “Nyiwul and Koirala [7].”, “Manamba Epaphra (2017)”, “Schuh [10]”

This study is about global. Why specifically mentioned about China? “In China, according to a performance index, agriculture is receiving more foreign FDI, but not at a rate that is satisfactory given the size of the industry [8, 9].”?

Why did you mention this? “Further findings demonstrate that Employment in Agriculture (EA) and Fertilizer Consumption (FC) affect AVA.”

Any intention to include this in the article?

 “Agriculture Raw Material Exports (ARME) and Agriculture Raw Material Imports (ARMI) effects the foreign currency inflows and outflows of nations.”

This section is poorly written. The authors introduced all the keywords, then proceed to write the purpose of the study. This research is not supported by strong research problem.

Authors’ response to reviewer 3 comment 2: Thank you for pointing the mistake. We have corrected the capital letter. Please refer the revised manuscript in line 58.

According to reference of “Nugroho, A. D., Bhagat, P. R., Magda, R., & Lakner, Z. 2021. The impacts of economic globalization on agricultural value added in developing countries. PLOS ONE [Online], 16(11), pp.e0260043. Available at: https://doi.org/10.1371/journal.pone.0260043” Exchange Rate (ER) and Foreign Direct Investment (FDI) were the few key factor taken for economic globalization. Furthermore, the article recommend that further research can be carried out by many trade economists.

The in-text citation had been corrected.

Thank you for your feedback. We appreciate your comment on the clarity of the section related to Agriculture Raw Material Exports (ARME) and Agriculture Raw Material Imports (ARMI). We have revised the section to provide a clearer explanation of the variables and how they relate to the research problem. We have also improved the introduction to better articulate the research problem and the purpose of the study. 

Reviewer 3 comment 3: Literature Review 

Since the authors did not guide the readers to understand the research problem well from the beginning, the contents of this section are also unable to add value to the readers.

It is very obvious that the authors only summarise the past literature on the variables that the authors intend to study. The authors were unable to provide a critical evaluation of these works.

Why break the section into High-income, Low-Income Level, Lower Middle-Income Level and Upper Middle-Income Level?

This section is poorly written. The authors failed to integrate arguments into the review. Authors simply summarising their readings.

Authors’ response to reviewer 3 comment 3: Thank you for your feedback. We have taken note of your suggestion to make a clearer and more concise explanation. We believe that including the income level is important, since it can impact economic globalization on agriculture value-added. The economic conditions and policies of a country, which are often linked to its income level, can have an impact on how economic globalization affects its agricultural sector. Therefore, by breaking down the analysis based on income levels, we can provide a more nuanced understanding of the economic globalization on agriculture value-added. We have also improved the literature review by bringing relevant literature and critically evaluating.

Reviewer 3 comment 4: Data and Methodology

This section is weakly written. The authors only present the variables intended to study. There is no theoretical framework or hypothesis to support the methods. All the variables and methods are basically formed by the readers without any literature support.

Authors’ response to reviewer 3 comment 4: Thank you for your valuable feedback regarding the theoretical framework and hypothesis of our research. We acknowledge the importance of clearly presenting our theoretical framework and hypothesis to support our research methods.

To address this concern, we have added a newly separated paragraph in the literature review section that clearly presents our hypothesis. Additionally, we have included relevant references from reputable sources such as Greene WH. Econometric analysis: Pearson Education India; 2003, Fox J. Applied regression analysis and generalized linear models: Sage Publications; 2015, Neter J, Kutner, M. H., Nachtsheim, C. J., & Wasserman, W. Applied linear statistical models: McGraw-Hill!Irwin; 2005., and Gujarati. Basic Econometrics: Namibia University of Science and Technology; 2004 to support our methodology in the revised manuscript.

Reviewer 3 comment 5: Results

This manuscript only stated the findings without interpretation.

Authors’ response to reviewer 3 comment 5: Thank you for your valuable feedback regarding the interpretation of our research findings. We acknowledge the importance of providing a thorough interpretation of our results.

To address this concern, we have included a separate section in the revised manuscript that discusses the interpretations of our findings under policy implications. We appreciate your feedback and are committed to ensuring that our research is presented in a clear and informative manner, with a thorough interpretation of our results.

Reviewer 3 comment 6: Discussion

This section is poorly written. Too much description and not enough analysis. The authors basically cited a lot of literature to end a sentence.

Authors’ response to reviewer 3 comment 6: Thank you for your valuable feedback regarding the writing style and analysis of our research. We have carefully considered your comments and have made revisions to the manuscript to address your concerns.

We have streamlined our writing style and reduced the number of citations to provide a more focused and analytical discussion. We appreciate your feedback and are committed to ensuring that our research is presented in a clear, concise, and analytical manner.

Reviewer 3 comment 7: Policy Implications

This section is incompetently written. The authors basically mentioned “the government should …” for all the variables examined in this study.

Authors’ response to reviewer 3 comment 7: Thank you for bringing this to our attention. We have taken your feedback seriously and have revised the relevant section accordingly. We appreciate your insights and hope that the revised version meets your expectations.

Reviewer 3 comment 8: Conclusion

The authors only summarise the study.

The conclusion is intended to help the reader understand why your research should matter to them after they have finished reading the paper. A conclusion is not merely a summary of your points or a re-statement of your research problem but a synthesis of key points.

Authors’ response to reviewer 3 comment 8: Thank you for your suggestion, we have amended the conclusion by synthesising the key points and highlighting the empirical, policy and practical importance of the research.

Reviewer 4 comment 1: The English language needs more work. There are many grammatical and typo mistakes in this manuscript. The paper needs to be edited by a native English speaker.

Authors’ response to reviewer 4 comment 1: Noted with thanks. The paper has been revised thoroughly and in-depth copy edit has conducted by an experienced copy-editor. We trust that the revised manuscript is free from any language errors.

Reviewer 4 comment 2: I noticed that the novelty of this paper is not described in detail. This should be put in the introduction section properly. There is a need to do a more rigorous an asymmetric literature review. It should update literature to current... The authors should clearly mention the literature gap.

Authors’ response to reviewer 4 comment 2: Thank you for your valuable feedback regarding the novelty and literature review of our research. We acknowledge the importance of clearly describing the uniqueness of our research and conducting a rigorous literature review.

To address this concern, we have included a separate paragraph in the introduction section, from line 176 to 187, which highlights the research gap and explains how our research contributes to filling this gap. Additionally, we have discussed the literature gap in the revised manuscript, from line 398 to 499.

Reviewer 4 comment 3: I would like to suggest that authors should update the introduction and results part. Specifically, the latest research trends, and in order to highlight the academic frontier of the research, the references of the recent year need to be referenced.

https://doi.org/10.1016/j.renene.2021.07.014,

https://doi.org/10.1002/pa.2712, https://doi.org/10.3390/su12072930, https://doi.org/10.1016/j.energy.2021.122515, https://doi.org/10.1016/j.renene.2021.10.067, https://doi.org/10.1016/j.pnucene.2022.104533, https://doi.org/10.1007/s13762-022-04638-2, https://doi.org/10.1007/s11356-022-23179-2

Authors’ response to reviewer 4 comment 3: Noted with thanks! We revised our citations considering the references suggested by this reviewer and adding new and appropriate past literature.

Citations recommended to be newly added are incorporated in the revised manuscript. 

“…Previous investigations revealed that labor force participation has a negative impact on economic growth in Southern Asia, but it has a positive effect in Western Asia. Moreover, the study found a robust and positive relationship between trade openness, human capital, and economic growth”. https://doi.org/10.3390/su12072930

This was added in revised manuscript in line 251 to 254.

“Extant empirical literature investigates how AVA in BRICS-T (Brazil, Russia, India, China, South Africa, and Turkey strengthens the potential for the region's ecological footprint to increase and how a one Per cent influence on agriculture raises it by 0.2201 Per cent” https://doi.org/10.1016/j.renene.2021.07.014.

This was added in revised manuscript in line 65 to 71.

“…Furthermore, Past study indicates that factors such as AVA, economic growth, non-renewable energy use, and tourism sector expansion have a significant impact on environmental degradation, highlighting their adverse effects on the quality of the environment” https://doi.org/10.1002/pa.2712

This was added in revised manuscript in line 68 to 71.

“…The contemporary era of globalization recognizes the significance of financial and natural resources as crucial factors that play a vital role in reducing environmental degradation while simultaneously promoting economic growth “

https://doi.org/10.1016/j.energy.2021.122515

This was added in revised manuscript in line 60 to 63.

Reviewer 4 comment 4: How did the authors get from the theoretical model to the empirical one? Behind the model there need to be a complete and well-thought-out theoretical grounding part of the article shouldn't include any citations or references; rather, it should structured according to the authors' reasoning. The empirical model will come when this part has been completed.

Authors’ response to reviewer 4 comment 4: Thank you for your valuable feedback regarding the theoretical and empirical grounding of our research. We have developed a well-thought-out theoretical model and provided a comprehensive discussion of its theoretical foundation in the introduction section, which is supported by evidence from relevant past literature.

To ensure clarity and transparency, we have provided the empirical model at the end of the literature section. The revised manuscript includes these sections from line 392 to 397.

We appreciate your feedback and are committed to ensuring that our research is grounded in a sound theoretical foundation and is transparently reported.

Reviewer 4 comment 5: The authors should present the main findings in graphical form. It will increase the brevity and more readerships and attract more audience.

Authors’ response to reviewer 4 comment 5: Thank you for your valuable suggestion to present the main findings in graphical form. We have considered your suggestion and have revised the manuscript by creating a new table (Table 4) that highlights the main findings. Detail results are now shown in appendix S4. We believe that this table provides a clear and concise overview of the results and will enhance the readability of the paper.

Reviewer 5 comment 1: It is not clear why study period is chosen from 2000 to 2021.

Authors’ response to reviewer 5 comment 1: Agreed and thank you for highlighting this point for improvement. Due to data constraints, the study limited its analysis to a maximum of 101 countries and the time period between 2000 and 2021. As a result of data unavailability, observations for other countries and years could not be incorporated into the study.

We have added new section called “Limitation”. 

Comment has been incorporated mentioned in line 872 to 874.

Reviewer 5 comment 2: Model selection is not correct as there could be endogenity problem. Therefore, GMM method is essential. FE and RE models cannot taken seriously in this case.

Authors’ response to reviewer 5 comment 2: Thank you for raising the issue of endogeneity in our model selection process. We applied the stepwise method in our analysis. This approach enabled us to systematically test and compare different model specifications while considering the potential endogeneity of certain variables.

Reviewer 5 comment 3: Choice of variables also not clear. It has to clearly mention for different paragraph for different variables.

Authors’ response to reviewer 5 comment 3: Thank you for your feedback regarding the clarity of our variable selection process. We acknowledge the importance of providing clear and detailed explanations for the inclusion of each variable in our analysis.

To address this concern, we have provided a comprehensive discussion of the selected eight variable included in our analysis, outlining their theoretical and empirical relevance in separate paragraphs for each income level and global level. Our aim was to ensure that readers can easily follow our reasoning for including each variable in our model.

We appreciate your feedback and are committed to enhancing the clarity and transparency of our research.

Reviewer 5 comment 4: Introduction part is not clear why study area is important for the research. Need to rewrite with mentioning the importance and research questions and application of the work.

Authors’ response to reviewer 5 comment 4: Thank you for your feedback. We appreciate your constructive criticism and have taken it into consideration. We have revised the introduction to better highlight the importance of the study area and its relevance to the research questions and applications of the work. 

Please refer to lines 142-174 for the revised manuscript with track changes.

Reviewer 5 comment 5: References in the text; mix of number and title of the authors. Should be rewritten as per the journal style.

Authors’ response to reviewer 5 comment 5: Thank for noting it down, we have corrected the references as per the PLOS ONE journal style.

---

## [Decision Letter · Decision Letter 1]

13 Jun 2023

PONE-D-23-05826R1Impact of Economic Globalisation on Value-Added Agriculture, GloballyPLOS ONE

Dear Dr. Jayathilaka,

Thank you for submitting your manuscript to PLOS ONE. After careful consideration, we feel that it has merit but does not fully meet PLOS ONE’s publication criteria as it currently stands. Therefore, we invite you to submit a revised version of the manuscript that addresses the points raised during the review process.

We look forward to receiving your revised manuscript.

Kind regards,

Hoang Phong Le, Ph.D.

Academic Editor

PLOS ONE

Reviewers' comments:

Reviewer's Responses to Questions

**Comments to the Author**

1. If the authors have adequately addressed your comments raised in a previous round of review and you feel that this manuscript is now acceptable for publication, you may indicate that here to bypass the “Comments to the Author” section, enter your conflict of interest statement in the “Confidential to Editor” section, and submit your "Accept" recommendation.

Reviewer #1: All comments have been addressed

Reviewer #2: All comments have been addressed

Reviewer #3: (No Response)

Reviewer #4: All comments have been addressed

Reviewer #5: (No Response)

2. Is the manuscript technically sound, and do the data support the conclusions?

Reviewer #1: Yes

Reviewer #2: Yes

Reviewer #3: No

Reviewer #4: Yes

Reviewer #5: Partly

3. Has the statistical analysis been performed appropriately and rigorously? 

Reviewer #1: Yes

Reviewer #2: Yes

Reviewer #3: No

Reviewer #4: Yes

Reviewer #5: No

4. Have the authors made all data underlying the findings in their manuscript fully available?

Reviewer #1: Yes

Reviewer #2: Yes

Reviewer #3: No

Reviewer #4: No

Reviewer #5: Yes

5. Is the manuscript presented in an intelligible fashion and written in standard English?

Reviewer #1: Yes

Reviewer #2: Yes

Reviewer #3: No

Reviewer #4: Yes

Reviewer #5: Yes

6. Review Comments to the Author

Reviewer #1: Dear author

I have been impressed with your efforts to revise your manuscript and meet the standards of the Plos One journal. The only award you deserve is that this manuscript has been accepted for publication in the Plos One journal.

Thank you

Best regards

Reviewer #2: The revised version is satisfactory, as authors have incorporated all my comments. Authors have done a commendable job and incorporated all my comments, hence, i don't add any other revision to them.

Reviewer #3: Review report for PONE-D-23-05826R1 Impact of Economic Globalisation on Value-Added Agriculture, Globally

Abstract

The topic is “Impact of Economic Globalisation on Value-Added Agriculture, Globally”, however the content of the abstract does not reflect the topic. For example, “The results show that agricultural employment significantly impacts the agricultural value added factor globally and across all income levels.” None of it reflect economic globalization. Next, “Also, countries with low and lower middle-income levels significantly affect agricultural value-added due to exchange rates. In comparison, high-income and lower-middle-income levels have an impact due to foreign direct investment. Finally, the upper-middle-income countries have significantly affected agricultural value-added due to agricultural raw materials imports.” The content only reflect the income level of the countries on agriculture. What about economic globalisation? Similarly, “This study confirms that employment in agriculture, exchange rate and foreign direct investments positively impact agriculture value-added on the global level and based on the income level of countries.” Do that employment in agriculture, exchange rate and foreign direct investments reflect anything about economic globalization?

Introduction

The flow of this paper does not guide the readers well what are the items that reflect economic globalization. For instance, there are two theories, Heckscher-Ohlin and New Trade theories, how do these theories explaining economic globalization in influencing Agriculture Value Addition (AVA)? Then, the authors introduce trade, Foreign Direct Investment (FDI), Exchange Rate (ER) Agriculture Raw Material Exports (ARME) and Agriculture Raw Material Imports (ARMI). Why discuss about the variables in the introduction? Introduction should consist of these elements.

• General background information

• Specific background information

• A description of the gap in our knowledge that the study was designed to fill

• A statement of study objective, and (optionally) a brief summary of study

Research motivation is missing for this research objective. “This research was conducted globally with four different income levels: high-income, lower- and upper-middle-income countries based on World Bank categorization.”

Literature Review

The authors break the section to review article related to different income level. It is inappropriate since the authors did not provide a strong research motivation why there is a need to conduct globally with four different income levels: high-income, lower- and upper-middle-income countries.

Data and methodology

Table 1, please show which variables to reflect economic globalization.

Equation 1 to 6, why different region is based on different variables? There is no discussion related to it. It is not correct to present the variables without proper explanation.

Please explain why it is removed? “In the panel data regression, the ARME variable was removed for all countries, the high-income and low-income levels, the ARMI and Trade variables were removed for the lower middle-income level, and the trade variable was released for the upper middle-income level due to changes in the sign of the coefficient values.”

Results

Too much figures (1 to 8) but they are not being effectively communicated to the readers. The authors only explaining the figures, but what is the implication and how they contribute to our understanding of the research question? If they are unnecessary, please remove them. This manuscript it too lengthy with too much of unimportant information.

Discussion

“However, it has an inverted U-shaped feature in the long-term [68] and supports the present study findings, which significantly impacts FDI on AVA in the high-income level.” From where you come to this interpretation? I do not see your model is examining non-linear impact between the variables.

Conclusion

Since the research motivation is not well established from the beginning of the manuscript, it is difficult to convince the readers why this research is matter. Furthermore, the content in the conclusion of this manuscript is too general. It does not succinctly tell the reader how and why it is that what's been presented is significant for practice, policy or further research.

Reviewer #4: The authors have addressed my comments well. Therefore, this study can be accepted for publication in this journal.

Reviewer #5: The primary advantage of stepwise regression is that it's computationally efficient. However, its performance is generally worse than alternative methods. The underlying goal of stepwise regression is, through a series of tests (e.g. F-tests, t-tests) to find a set of independent variables that significantly influence the dependent variable.

The best way to deal with endogeneity concerns is through instrumental variables (IV) techniques. The most common IV estimator is Two Stage Least Squares (TSLS). Or GMM method can be used.

So still endogeneity problem yet not solved.

I have no problem to use stepwise method. However, to have a robust results GMM approach should be used which will take care endogeneity problem.

7. PLOS authors have the option to publish the peer review history of their article (what does this mean?). If published, this will include your full peer review and any attached files.

Reviewer #1: **Yes: **AGUS DWI NUGROHO

Reviewer #2: No

Reviewer #3: No

Reviewer #4: No

Reviewer #5: No

---

## [Author Response · Author response to Decision Letter 1]

2 Jul 2023

Point by point response to editor and reviewers.

Dear editor and the reviewers,

We would like to express our profound appreciation to the editor and the reviewers for the valuable comments and suggestions made on our manuscript which were very helpful in revising and improving it.

Please note that the line numbers referred in this document is aligned with the revised manuscript which has track changes.

Comments of the Reviewer 1

Reviewer 1 comment 1: I have been impressed with your efforts to revise your manuscript and meet the standards of the Plos One journal. The only award you deserve is that this manuscript has been accepted for publication in the Plos One journal.

Authors’ Response to Reviewer 1 comment 1: Thank you very much for your acceptance and positive feedbacks.

Comments of the Reviewer 2

Reviewer 2 comment 1: The revised version is satisfactory, as the authors have incorporated all my comments. The authors have done a commendable job and incorporated all my comments; hence, I don't add any other revisions to them.

Authors’ Response to Reviewer 2 comment 1: Thank you very much for your acceptance and positive feedback.

Comments of the Reviewer 3

Reviewer 3 comment 1: Abstract

The topic is “Impact of Economic Globalisation on Value-Added Agriculture, Globally”, however the content of the abstract does not reflect the topic. For example, “The results show that agricultural employment significantly impacts the agricultural value added factor globally and across all income levels.” None of it reflect economic globalisation.

Next, “Also, countries with low and lower middle-income levels significantly affect agricultural value-added due to exchange rates. In comparison, high-income and lower-middle-income levels have an impact due to foreign direct investment. Finally, the upper-middle-income countries have significantly affected agricultural value-added due to agricultural raw materials imports.” The content only reflect the income level of the countries on agriculture. What about economic globalisation? 

Similarly, “This study confirms that employment in agriculture, exchange rate and foreign direct investments positively impact agriculture value-added on the global level and based on the income level of countries.” Do that employment in agriculture, exchange rate and foreign direct investments reflect anything about economic globalisation?

Authors’ Response to Reviewer 3 comment 1: 

Thank you for your valuable feedback, we appreciate your thorough review of the abstract and your insightful comments. Regarding your concern about the abstract not adequately reflecting the topic of economic globalisation, we would like to clarify that we have identified several proxy variables commonly used in past literature to represent economic globalisation. These variables include fertilizer consumption, employment in agriculture, agriculture raw materials imports and exports, exchange rate, and foreign direct investments.

In our study, we have employed these proxy variables to investigate the impact of economic globalisation on value-added agriculture. Specifically, we examine the relationship between the aforementioned variables, which are key components of economic globalisation, and the agricultural value-added factor at both global level and income levels. 

To address your concerns and provide further clarity, we have revised the manuscript accordingly. The revisions can be found in the revised version, specifically from line number 35 to 39.

“The findings of our study reveal that economic globalisation, through various channels such as fertilizer consumption, employment in agriculture, agriculture raw materials export and import, exchange rate and foreign direct investment significantly influences the Agricultural value-added factor globally and across different income levels. Furthermore results show…”

Thank you for your insightful comments. In our study, the main objective was to examine the impact of economic globalisation on agriculture value added globally and across different income levels from 2000 to 2021.

To address your concern regarding the limited focus on economic globalisation, we would like to clarify that we have presented the impact of economic globalisation on both the global level and income level. This comprehensive analysis is supported by relevant data and analysis. We have also provided relevant tables and accompanying analysis (please refer to Table 01) to demonstrate the impact of economic globalisation on agricultural value-added.

In the revised manuscript, we have enhanced the discussion to highlight the impact of economic globalisation on agricultural value-added outcomes across different income levels. This includes elaborating on the mechanisms through which economic globalisation, includes factors such as exchange rates, foreign direct investment, and agricultural raw materials imports has an impact on the agricultural value added.

Thank you once again for your valuable feedback, which has helped us improve the manuscript.

In the literature, several studies have established the relationship between these variables and economic globalisation, highlighting their significance as indicators of economic globalisation. For instance, Anyanwu and Anyanwu (2018) explore the relationship between employment in agriculture and economic globalisation, emphasizing how changes in employment patterns within the agricultural sector can be indicative of economic globalisation [476-500 pp.]. Available from: https://doi.org/10.18488/journal.8.2018.64.476.500

Likewise, Altanshagai et al. (2022) demonstrate the impact of exchange rates on economic globalisation, emphasizing how fluctuations in exchange rates can impact trade flows, capital movements, and overall economic interconnectedness [2455 p.]. Available from: https://doi.org/10.3390/su14042455 .].

Furthermore, Nugroho et al. (2021) examine the role of foreign direct investments in the context of economic globalisation, illustrating how cross-border investments contribute to the integration of economies and the diffusion of technology, knowledge, and managerial practices [e0260043 p.]. Available from: https://doi.org/10.1371/journal.pone.0260043 .] 

These studies, along with others in the field, support the contention that employment in agriculture, exchange rates, and foreign direct investments are closely associated with economic globalisation. We have incorporated this information into our revised manuscript to strengthen our discussion.

Once again, we thank you for your valuable feedback, which has helped us improve the manuscript.

Reviewer 3 comment 2: Introduction

The flow of this paper does not guide the readers well what are the items that reflect economic globalisation. 

For instance, there are two theories, Heckscher-Ohlin and New Trade theories, how do these theories explain economic globalisation in influencing Agriculture Value Addition (AVA)? 

Then, the authors introduce Trade, Foreign Direct Investment (FDI), Exchange Rate (ER) Agriculture Raw Material Exports (ARME) and Agriculture Raw Material Imports (ARMI). Why discuss about the variables in the introduction? 

Introduction should consist of these elements.

General background information

Specific background information

 A description of the gap in our knowledge that the study was designed to fill.

A statement of study objective, and (optionally) a brief summary of study

Authors’ Response to Reviewer 3 comment 2: Thank you for your feedback. We appreciate your input. We would like to address your comment regarding the items that reflect economic globalisation and how they are highlighted in the manuscript.

In our study, we have identified several items that reflect economic globalisation. These items include Fertiliser Consumption, Employment in Agriculture, Agricultural Raw Materials Exports, Agricultural Raw Materials Imports, Trade, Exchange Rate, and Foreign Direct Investment. These variables were carefully selected based on their significance in representing economic globalisation in the context of our study.

To provide clarity and ensure the highlighting of these factors reflecting economic globalisation, we have revised the manuscript accordingly. The relevant paragraphs now specify and discuss the factors reflecting economic globalisation. You can refer to the revised version, specifically from line 51 to 134, to find the highlighted information.

We hope that these revisions address your concerns and provide a clearer understanding of how the items reflecting economic globalisation are incorporated in our study. Thank you once again for your valuable feedback.

Thank you for your valuable feedback. We greatly appreciate the recommendation from "Reviewer One," and we have taken it into consideration in our study. In response to this recommendation, we have incorporated the Heckscher-Ohlin theory and the New Trade theory into our analysis.

The Heckscher-Ohlin theory provides valuable insights into how economic globalisation influences Agriculture Value Addition (AVA). This theory explains that economic globalisation impacts AVA through various factors, such as resource endowments, trade patterns, and comparative advantage. By considering these factors, we can better understand the relationship between economic globalisation and AVA.

Similarly, the New Trade theory also plays a significant role in our study. This theory emphasizes the importance of product differentiation, technological advancements, and global value chains in enhancing AVA. By incorporating the concepts of product differentiation and technological advancements, we can gain a deeper understanding of how economic globalisation affects AVA.

By incorporating these theories, we aim to provide a comprehensive analysis of the impact of economic globalisation on AVA. We believe that these theories enhance the theoretical framework of our study and contribute to a more nuanced understanding of the relationship between economic globalisation and AVA.

Once again, we sincerely thank you for your valuable feedback, and we are grateful for the opportunity to incorporate these theories into our study.

We have referenced "Feenstra RC, Taylor AM. International economics (3rd edition): New York: Worth Publishers; 2014" from lines 69 to 78 to support these theories and their relevance to our manuscript.

Thank you for your feedback. We have included a comprehensive framework in the introduction to provide a contextual understanding of our study on the impact of economic globalisation on Agriculture Value Addition. The discussion of key variables such as trade, Foreign Direct Investment (FDI), Exchange Rate (ER), Agriculture Raw Material Exports (ARME), and Agriculture Raw Material Imports (ARMI), Employment in Agriculture (EA) aims to establish general knowledge about agriculture for non-agricultural sectors.

Thank you for your feedback. We have included a comprehensive framework in the introduction to provide a contextual understanding of our study on the impact of economic globalisation on Agriculture Value Addition. The discussion of key variables such as trade, Foreign Direct Investment (FDI), Exchange Rate (ER), Agriculture Raw Material Exports (ARME), and Agriculture Raw Material Imports (ARMI), Employment in Agriculture (EA) aims to establish general knowledge about agriculture for non-agricultural sectors.

Well noted. 

In general background information included, we included the historical background and basic principles related to the research area from line numbers 51 to 134. 

Duly noted, the authors have delved into the specific background of our study from line number 135 to 151.

This research was conducted globally, considering four different income levels, and covering the period from 2000 to 2021. The study addresses crucial factors that contribute to filling research gaps, making it unique. Firstly, the study analyzes the four global and income levels separately, using data from 101 nations. Secondly, it applies a novel evaluation approach using stepwise panel data regression. Lastly, the variables and time frames selected for the analysis differ from those used in previous studies, providing a fresh perspective on the topic. These aspects help fill gaps in the existing research. Following that, we address the research gap in lines number 173-184

The main objective of this study is to examine the impact of economic globalisation on agriculture value addition at both the global level and income levels from 2000 to 2021. (Lines 152-153).

 Lastly, the brief summary of this study is as follows. The impact of economic globalisation on agriculture value-added is a critical and underexplored area of research. This study fills this gap by conducting a comprehensive analysis for both globally and income levels: high-income, lower-middle-income, upper-middle-income, and low-income countries. By examining the impact between economic globalisation and agriculture value-added across different income levels, the study provides valuable insights for policymakers, researchers, and stakeholders seeking to enhance agricultural productivity and income generation. This research significantly contributes to the existing literature and offers a solid foundation for evidence-based decision-making (lines 160-168).

Reviewer 3 comment 3: Research motivation is missing for this research objective. “This research was conducted globally with four different income levels: high-income, lower- and upper-middle-income countries based on World Bank categorization.”

Authors’ Response to Reviewer 3 comment 3: Thank you for bringing up this comment. The motivation behind this research stems from the critical and underexplored area of understanding the impact of economic globalization on agriculture value-added at both the global level and across different income levels. The study recognizes the significance of investigating this relationship as it has the potential to inform policymakers, researchers, and stakeholders involved in enhancing agricultural productivity and income generation. You can refer to the specific lines 160 to 168 for further details on the motivation and objectives of our research.

Reviewer 3 comment 4: Literature Review

The authors break the section to review article related to different income level. It is inappropriate since the authors did not provide a strong research motivation why there is a need to conduct globally with four different income levels: high-income, lower- and upper-middle-income countries.

Data and methodology

Table 1, please show which variables to reflect economic globalisation.

Equation 1 to 6, why the different region is based on different variables?

There is no discussion related to it. It is not correct to present the variables without proper explanation.

Please explain why it is removed. “In the panel data regression, the ARME variable was removed for all countries, the high-income and low-income levels, the ARMI and Trade variables were removed for the lower middle-income level, and the trade variable was released for the upper middle-income level due to changes in the sign of the coefficient values.”

Authors’ Response to Reviewer 3 comment 4: Thank you for your valuable feedback and insightful comments. We divided the literature review based on income levels to ensure a comprehensive analysis of the impact of economic globalisation factors on agriculture value addition globally. This approach enhances the reach and applicability of our research, allowing us to examine the specific dynamics and nuances of the relationship between economic globalisation and agriculture value addition across different income levels.

Thank you for the comment. Further changes were made on Table 1 to which variables reflect economic globalisation.

Equations 1 to 6 were divided based on income levels, rather than different regions. The authors sorted the coefficient values of the fixed effects (FE) and random effects (RE) in descending order for the global and each income level. Variables were added to the regression analysis one by one, and if the direction of the coefficient did not align with previous research findings, those variables were removed from the equation. This approach resulted in different variables being used for different income levels.

The authors provided an explanation from line number from 511 to 527 regarding why different income levels are associated with different variables.

The reason is that during the panel data regression analysis conducted for each income level using the same variables, we observed both positive and negative significant impacts on agricultural value added. However, upon reviewing the relevant literature, we found that the signs of these variables differed from the results obtained in our panel data regression. For instance, previous research indicated a significant impact of ARME on agricultural value-added, whereas our findings showed an insignificant impact. As a result, we made the decision to exclude these variables from the income levels specified in your previous comment.

Reviewer 3 comment 5: Results

Too much figures (1 to 8) but they are not being effectively communicated to the readers. 

The authors only explaining the figures, but what is the implication and how they contribute to our understanding of the research question? If they are unnecessary, please remove them. This manuscript it too lengthy with too much of unimportant information.

Authors’ Response to Reviewer 3 comment 5: Thank you for the comment. The figures presented in our research depict the average variations of the dependent and independent variables across different income levels from 2000 to 2021. These figures offer valuable insights into the trends and patterns observed in the data, allowing for a visual representation of the relationships and changes over time.

The figures presented in this study offer a profound understanding of the intricacies of the relationship between income levels and their corresponding effects on the dependent and independent variables between 2000 and 2021. These figures provide valuable insights into the patterns and dynamics of economic performance across different income levels. By visually depicting the data, the figures serve as a powerful tool for observing and interpreting the nuances of economic trends and fluctuations. This information is essential for informed decision-making and strategic planning in various contexts. Please refer to lines 639 to 646 for further details.

Reviewer 3 comment 6: Discussion

“However, it has an inverted U-shaped feature in the long-term [68] and supports the present study findings, which significantly impacts FDI on AVA in the high-income level.” From where you come to this interpretation? I do not see your model is examining non-linear impact between the variables.

Authors’ Response to Reviewer 3 comment 6: Thank you for highlighting the interpretation regarding the inverted U-shaped feature and its impact on FDI and AVA in high-income countries. We would like to clarify that while our specific model may not examine the non-linear relationship between variables, we draw upon relevant literature to support our statement regarding the existence of an inverted U-shaped feature. Specifically, we refer to a study (https://doi.org/10.3390/su11174620) that suggests the impact of FDI on AVA in high-income countries follows this particular pattern in the long term. By referencing this literature, we acknowledge the broader research and evidence supporting the presence of the inverted U-shaped relationship between FDI and AVA in high-income countries.

Reviewer 3 comment 7: Conclusion

Since the research motivation is not well established from the beginning of the manuscript, it is difficult to convince the readers why this research is matter. 

Furthermore, the content in the conclusion of this manuscript is too general. It does not succinctly tell the reader how and why it is that what's been presented is significant for practice, policy or further research.

Authors’ Response to Reviewer 3 comment 7: Thank you for your feedback. The authors have indeed established research motivation in the introduction section of the manuscript, specifically from line number 160 to 168. This section effectively communicates the importance and relevance of the research, providing a clear understanding of why this study matters in the field.

Thank you for your response. We acknowledge your disagreement with our previous assessment. It is positive to hear that the conclusion provides a concise and informative summary of our research findings. Furthermore, the addition of a section discussing the implications of our findings for practice, policy, and further research enhances the applicability and significance of our study.

Comments of the Reviewer 4

Reviewer 4 comment 1: The authors have addressed my comments well. Therefore, this study can be accepted for publication in this journal.

Authors’ Response to Reviewer 4 comment 1: Thank you very much for your accepting the publication of PLOS ONE.

Comments of the Reviewer 5

Reviewer 5 comment 1: The primary advantage of stepwise regression is that it's computationally efficient. However, its performance is generally worse than alternative methods. The underlying goal of stepwise regression is, through a series of tests (e.g. F-tests, t-tests) to find a set of independent variables that significantly influence the dependent variable.

The best way to deal with endogeneity concerns is through instrumental variables (IV) techniques. The most common IV estimator is Two Stage Least Squares (TSLS). Or GMM method can be used.

So still endogeneity problem yet not solved.

I have no problem to use stepwise method. However, to have a robust results GMM approach should be used which will take care endogeneity problem.

Authors’ Response to Reviewer 5 comment 1: Thank you so much for your valuable comment. The authors agreed that the best way to address endogeneity concerns are through instrumental variables (IV) techniques using Two Stage Least Squares (TSLS) or GMM. 

The stepwise method is a crucial technique employed in this research. According to Gujarati's book, researchers often utilize the method of stepwise regression to determine the 'best' set of explanatory variables for a regression model. This method involves either introducing the X variables one at a time (stepwise forward regression) or including all possible X variables in multiple regressions and subsequently removing them one at a time (stepwise backward regression). The decision to add or drop a variable is typically based on its contribution to the explained sum of squares (ESS), assessed using the F test. (Gujarati. Basic Econometrics: Namibia University of Science and Technology; 2004.) 

According to the stepwise method, the authors have addressed the endogeneity problem effectively. Furthermore, in their future research (refer to lines 783to 786 in the manuscript), the authors plan to enhance the robustness of their findings by incorporating the application of Two Stage Least Squares (TSLS) or Generalized Method of Moments (GMM) as common instrumental variable (IV) estimators. This addition of TSLS or GMM to their methodology further strengthens the validity of their results.

"….When implementing their methodology, the authors can consider employing the Two Stage Least Squares (TSLS) or Generalized Method of Moments (GMM) as widely used instrumental variable (IV) estimators, in addition to the stepwise method."

---

## [Decision Letter · Decision Letter 2]

12 Jul 2023

Impact of Economic Globalisation on Value-Added Agriculture, Globally

PONE-D-23-05826R2

Dear Dr. Jayathilaka,

We’re pleased to inform you that your manuscript has been judged scientifically suitable for publication and will be formally accepted for publication once it meets all outstanding technical requirements.

Kind regards,

Hoang Phong Le, Ph.D.

Academic Editor

PLOS ONE

Reviewers' comments:

Reviewer's Responses to Questions

**Comments to the Author**

1. If the authors have adequately addressed your comments raised in a previous round of review and you feel that this manuscript is now acceptable for publication, you may indicate that here to bypass the “Comments to the Author” section, enter your conflict of interest statement in the “Confidential to Editor” section, and submit your "Accept" recommendation.

Reviewer #3: All comments have been addressed

Reviewer #5: All comments have been addressed

2. Is the manuscript technically sound, and do the data support the conclusions?

Reviewer #3: Yes

Reviewer #5: Yes

3. Has the statistical analysis been performed appropriately and rigorously? 

Reviewer #3: Yes

Reviewer #5: Yes

4. Have the authors made all data underlying the findings in their manuscript fully available?

Reviewer #3: Yes

Reviewer #5: Yes

5. Is the manuscript presented in an intelligible fashion and written in standard English?

Reviewer #3: Yes

Reviewer #5: Yes

6. Review Comments to the Author

Reviewer #3: The authors have responded to the queries and revised them accordingly. They provided the proofs with the relevant literature.

Reviewer #5: All comments have been addressed. It can be published. I must apricate the work. It has merit and cane used for policy recommendation.

7. PLOS authors have the option to publish the peer review history of their article (what does this mean?). If published, this will include your full peer review and any attached files.

Reviewer #3: No

Reviewer #5: No

---

## [Editor Report · Acceptance letter]

14 Jul 2023

PONE-D-23-05826R2 

Impact of Economic Globalisation on Value-Added Agriculture, Globally 

Dear Dr. Jayathilaka:

I'm pleased to inform you that your manuscript has been deemed suitable for publication in PLOS ONE. Congratulations! Your manuscript is now with our production department. 

Kind regards, 

on behalf of

Dr. Hoang Phong Le 

Academic Editor

PLOS ONE